# Population coupling predicts the plasticity of stimulus responses in cortical circuits

**Yann Sweeney, Claudia Clopath\***

Department of Bioengineering, Imperial College London, London, United Kingdom

**Abstract** Some neurons have stimulus responses that are stable over days, whereas other neurons have highly plastic stimulus responses. Using a recurrent network model, we explore whether this could be due to an underlying diversity in their synaptic plasticity. We find that, in a network with diverse learning rates, neurons with fast rates are more coupled to population activity than neurons with slow rates. This plasticity-coupling link predicts that neurons with high population coupling exhibit more long-term stimulus response variability than neurons with low population coupling. We substantiate this prediction using recordings from the Allen Brain Observatory, finding that a neuron's population coupling is correlated with the plasticity of its orientation preference. Simulations of a simple perceptual learning task suggest a particular functional architecture: a stable 'backbone' of stimulus representation formed by neurons with low population coupling, on top of which lies a flexible substrate of neurons with high population coupling.

**\*For correspondence:**
c.clopath@imperial.ac.uk

**Competing interests:** The authors declare that no competing interests exist.

## Introduction

The brain encodes information about the external world via its neural activity. One aspect of such encoding is that neurons in sensory cortex often have a preferred stimulus which evokes a stronger response than other stimuli. These stimulus responses can change during learning or adaptation: if a particular stimulus feature is overexpressed within an environment, for example, more neurons will be recruited to encode this feature (*Sengpiel et al., 1999*). Advances in neural imaging techniques allow us to interrogate such changes by tracking stimulus responses of hundreds of neurons over many days in vivo (*Andermann, 2010*; *Mank et al., 2008*). These recordings reveal a substantial, and puzzling, variability in the long-term stability of responses in sensory cortex: some neurons retain highly stable preferences to specific stimuli, whereas the stimulus preference of other neurons change from day to day (*Ranson, 2017*; *Clopath et al., 2017*; *Poort et al., 2015*; *Lütcke et al., 2013*; *Rule et al., 2019*; *Rose et al., 2016*). The degree of stimulus response stability typically depends on brain region; whisking responses in mouse barrel cortex are highly plastic, whereas visual responses in mouse V1 are more stable but still exhibit fluctuations (*Clopath et al., 2017*; *Lütcke et al., 2013*). Moreover, it is possible to induce stimulus response plasticity through perturbations such as sensory deprivation (*Rose et al., 2016*), or to increase task-related stimulus response stability through rewarded learning (*Poort et al., 2015*).

Current theories which address the long-term variability of stimulus responses primarily ask how motor learning occurs with unstable representations (*Driscoll et al., 2017*; *Ajemian et al., 2013*; *Rokni et al., 2007*), or seek to explain it as a form of probabilistic sampling (*Kappel et al., 2017*; *Kappel et al., 2015*). Although the stability of neural representation is correlated with firing rate in hippocampal place cells (*Grosmark and Buzsáki, 2016*) and in visual cortex (*Ranson, 2017*), it is not known how cellular or network properties influence a neuron's stimulus response stability (*Clopath et al., 2017*). We are therefore lacking a theory of why some neurons' stimulus responses are more stable than others, and how this affects perception and learning. By investigating how

synaptic plasticity mediates stimulus response variability, we aim here to establish how this diversity of stimulus response stability emerges, and whether it is functionally relevant.

We propose that the observed diversity of stimulus response stability may be explained by a diversity of neurons' inherent plasticity (or learning rate) within a network. Consequently, we explore how diverse learning rates across neurons impact synaptic connectivity in a recurrent network model of mouse visual cortex. We find that neurons with fast learning rates exhibit more variability of their stimulus selectivity than neurons with slow learning rates. Intriguingly, we also find that fast neurons have higher population coupling, a measure of how correlated an individual neurons activity is with the rest of the population (*Okun et al., 2015*).

This unexpected plasticity-coupling link, in which more plastic neurons are also more coupled to the rest of the population, provides a mechanism for the diverse population coupling previously observed in sensory cortex (*Okun et al., 2015*). Moreover, the plasticity-coupling link predicts that neurons with high population coupling exhibit more long-term stimulus response variability than neurons with low population coupling. We substantiate this prediction with in vivo calcium imaging of mouse visual cortex from the Allen Brain Observatory (*Allen Brain Atlases and Data, 2016*), finding that a neuron is more likely to exhibit variability of its orientation preference if it has high population coupling.

Finally, we explore the functional implications of both diverse population coupling and diverse learning rates within our network model. We find that strong population coupling helps plastic neurons alter their stimulus preference during a simple perceptual learning task, but hinders the ability of stable neurons to provide an instructive signal for learning. The plasticity-coupling link exploits this dependence by ensuring that highly plastic neurons - the substrate for perceptual learning - are strongly coupled to the population, while less plastic neurons are weakly coupled and act as a stable 'backbone' of stimulus representation.

## Results

### A 'plasticity-coupling link' emerges in networks with diverse learning rates: fast neurons have higher population coupling than slow neurons

Our aim is to explore whether the diversity of stimulus response stability can be explained by a diversity of neurons' inherent plasticity (or learning rate) within a network. To this end, we use network simulations to characterise the impact of diverse learning rates on recurrent synaptic connectivity in sensory cortex.

We first explore the impact of diverse learning rates in a simple, fully connected network of rate neurons (*Figure 1A*, Materials and methods). Excitatory recurrent synapses in our network undergo Hebbian plasticity and synaptic scaling, while inhibitory synapses undergo homeostatic inhibitory plasticity (*Vogels et al., 2011*). Extending traditional models of Hebbian plasticity in which synaptic weight updates depend only on the correlation of pre- and post-synaptic activity, we introduce diversity by assigning either a fast or slow Hebbian learning rate ($\alpha$) to individual neurons. The learning rate is expressed postsynaptically, such that the synaptic input weights onto neurons with a large $\alpha$ are more plastic than those with a small $\alpha$ (*Equation 3*).

Each neuron receives feedforward input from 1 of 4 possible visual stimuli representing gratings of different orientations, and independent noise. The Hebbian plasticity rule potentiates connections between neurons which share the same feedforward stimulus preference, due to their coactivity. This drives the emergence of strong bidirectional connections amongst stimulus-specific groups of neurons, while the remaining non-specific connections weaken (*Figure 1B,C*; *Ko et al., 2013*; *Clopath et al., 2010*). Fast neurons develop these strong, specific connections sooner than slow neurons (*Figure 1B*, solid lines). However, the increased learning rate also leads to stronger synaptic weight fluctuations. These fluctuations occur both for synapses from neurons which share stimulus preference (specific connections) and for synapses from neurons which have different stimulus preference (non-specific connections). For slow neurons, in contrast, non-specific and specific connections tend towards either zero or the maximum synaptic weights respectively, remaining relatively stable after convergence (*Figure 1B*, black lines). This leads to connection specificity that is stronger and more stable compared with fast neurons (*Figure 1D*).

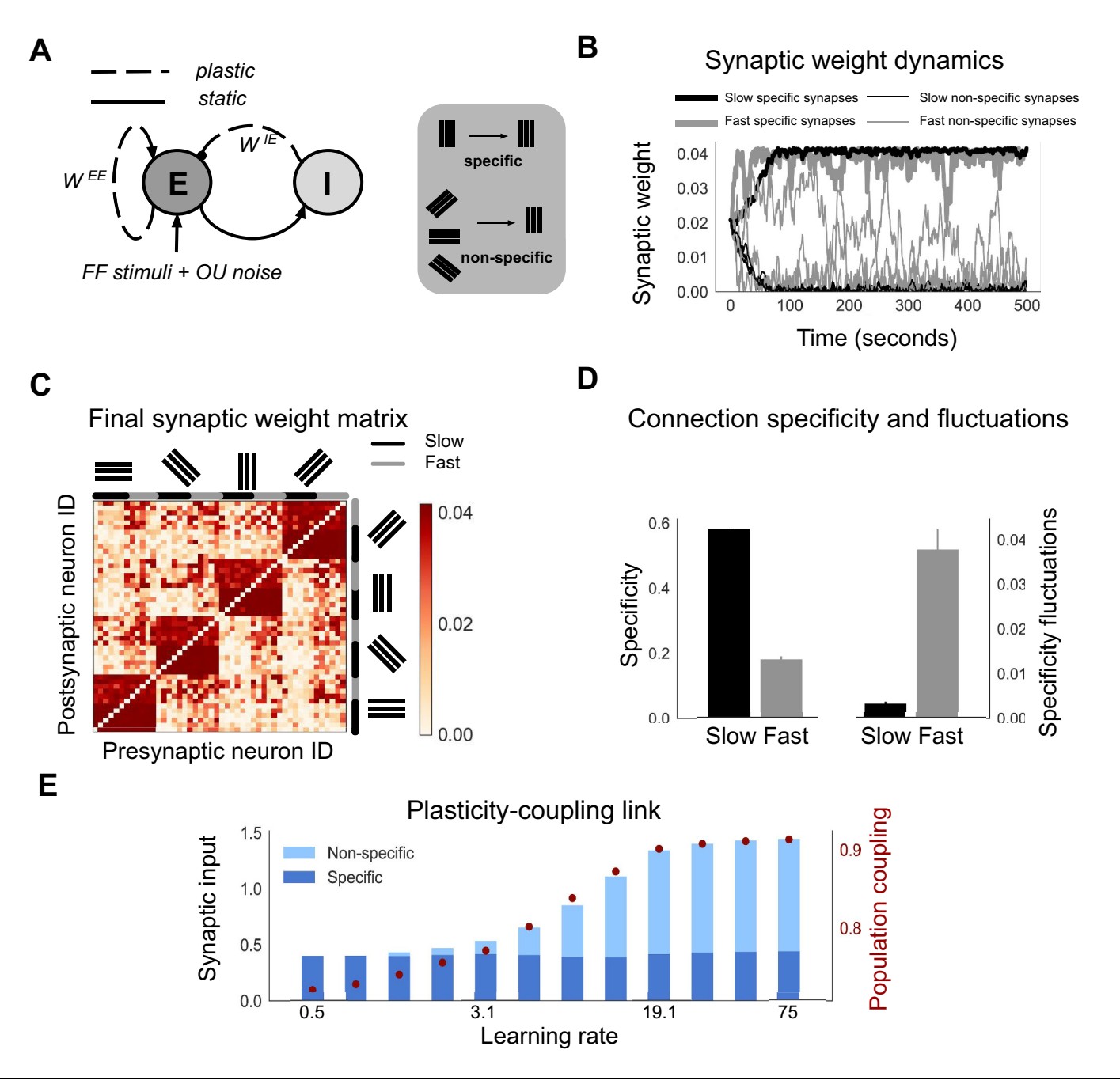

**Figure 1.** Neurons with fast learning rates develop more non-specific connections, and higher population coupling, than neurons with slow learning rates. (A) Connection diagram of the recurrent network model with excitatory (E) and inhibitory (I) neurons. Dashed lines denote plastic synapses and solid lines denote static synapses. (B) Synaptic weight dynamics during presentation of random sequences of stimuli to the network. Synaptic inputs onto slow neurons ($\alpha = 1$, gray) and onto fast neurons ($\alpha = 5$, black). Synapses between neurons which share the same feedforward stimulus preference (specific) have solid lines, and synapses between neurons which have different feedforward stimulus preference (non-specific) have dashed lines. (C) Excitatory synaptic weight matrix of the recurrent network after synaptic plasticity. Neuron IDs are organised by feedforward stimulus preference. For each of the four stimulus groups the first six neurons are slow ($\alpha = 1$) and the next six neurons are fast ($\alpha = 5$). (D) Connection specificity (ratio of specific to non-specific synaptic input strength) after synaptic plasticity for slow and fast neurons (left), and the standard deviation over time of the connection specificity for slow and fast neurons (right). (E) Amount of non-specific (light blue) and specific (dark blue) synaptic input for neurons in a network with diverse learning rates, as the learning rate of the postsynaptic neuron is varied along a logarithmic scale. Population coupling of neurons with different learning rates (red points).

The online version of this article includes the following figure supplement(s) for figure 1:

*Figure 1 continued on next page*

*Figure 1 continued*

**Figure supplement 1.** Plasticity-coupling link requires moderate noise and slow synaptic scaling.

The observed dependence of connection specificity on learning rate is conserved if, instead of just two values of $\alpha$ representing either fast or slow neurons, we simulate plasticity in a network of neurons with a diverse range of $\alpha$ (*Figure 1E*). Increasing $\alpha$ predominantly drives an increase in non-specific connections rather than a decrease in specific connections. This leads to an overall increase in the amount of synaptic input amongst neurons with high $\alpha$.

Population coupling is a recently characterised feature of neural activity which describes how correlated a neuron's activity is with the overall population activity, and which can be measured from calcium imaging recordings of neural activity (*Okun et al., 2015*). Since population coupling is correlated with the amount of local synaptic input in cortical networks (*Okun et al., 2015*), this measure could be a useful and experimentally observable proxy for the specificity of recurrent connectivity in our networks. We therefore investigate its suitability by measuring the population coupling of neurons in our network after synaptic plasticity (Materials and methods). Interestingly, population coupling increases with learning rate, closely following the dependence of non-specific connectivity on $\alpha$ (*Figure 1E*, red points).

The dependence of a neuron's population coupling on its learning rate $\alpha$, which we call a 'plasticity-coupling link', could provide a framework for relating the functional role of a neuron within a network to its dynamics. We therefore explore conditions necessary for this plasticity-coupling link by embedding a single plastic neuron within a static network and varying key model parameters (*Figure 1—figure supplement 1*). A strong plasticity-coupling link requires both moderate amounts of noise within the network and relatively slow synaptic scaling compared with Hebbian plasticity, in agreement with experimental data (*Turrigiano et al., 1998*). We next investigate whether this plasticity-coupling link is robustly observed in more biologically detailed networks.

## Diverse population coupling emerges in cortical networks with diverse learning rates

As the plasticity-coupling link is robustly observed in a fully-connected small network with simple stimulus responses, we next investigate i) whether the plasticity-coupling link is also present in larger networks which more accurately represent the synaptic connectivity and stimulus response properties observed in mouse visual cortex, and ii) whether the diverse population coupling observed in sensory cortex emerges simply by introducing diverse learning rates (*Okun et al., 2015*).

We explore this in a network of 250 excitatory neurons with randomly generated Gabor receptive fields. This network has been shown to reproduce receptive field correlations and synaptic weight statistics that are observed in mouse visual cortex (*Watanabe et al., 2016*; *Cossell et al., 2015*) (see Materials and methods section; Receptive-field based network model). We compare networks in which there is a uniform $\alpha$ across all neurons to networks with diverse $\alpha$.

Both networks with uniform $\alpha$ and networks with diverse $\alpha$ develop strong synaptic connections between neurons with similar receptive fields. There is, however, a broader range of summed synaptic inputs in diverse networks, when compared with uniform networks (*Figure 2A*). This occurs because the total excitatory synaptic input onto a neuron covaries with $\alpha$ in the diverse network (*Figure 2B*).

In agreement with our previous observations, the population coupling of a neuron is determined by its total excitatory synaptic input in networks with diverse $\alpha$ (*Figure 2C*, blue line. r = 0.29, p<1e-5, Spearman correlation). Diverse learning rates within a cortical network indeed lead to a broad distribution of population coupling, as observed by *Okun et al. (2015)*; *Figure 2D*, blue). Although the network with uniform $\alpha$ also exhibits some heterogeneity of population coupling, in this network a neuron's population coupling is not correlated with the amount of synaptic input it receives (*Figure 2C,D*, green. p=0.52, Spearman correlation). The network with diverse $\alpha$ exhibits population coupling which is both broadly distributed and correlated with synaptic input - in agreement with (*Okun et al., 2015*) - while the absence of correlation in the network with uniform $\alpha$ is in contrast with experiments which demonstrate a correlation between synaptic input and population coupling

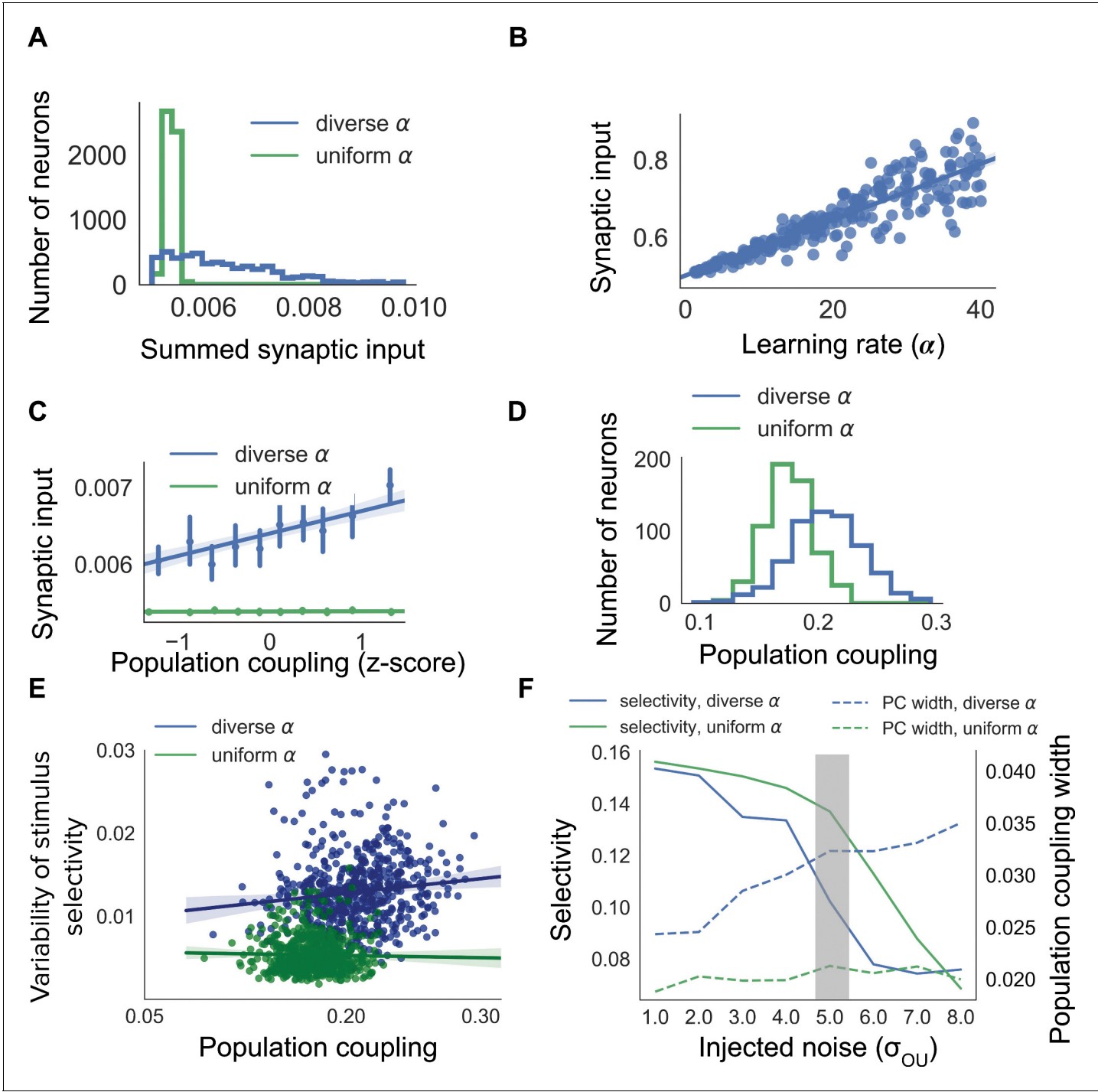

**Figure 2.** Diverse population coupling from diverse learning rates in a cortical network model. (**A**) Distribution of summed synaptic input onto each neuron in networks with diverse learning rates (blue), and networks with uniform learning rates (green) (**B**) Mean recurrent excitatory synaptic input received by a neuron correlates with its learning rate, $\alpha$. (**C**) The population coupling of a neuron is correlated with the amount of recurrent synaptic input it receives for the network with diverse learning rates (blue), as opposed to the network with uniform learning rates (green). Error bars for each bin show 95% confidence interval. Lines show linear regression fit for all datapoints (shaded coloured area indicates 95% confidence interval) (**D**) Diverse population coupling occurs in our recurrent network model. The population coupling distribution is wider for networks with diverse learning rates (blue) compared to networks with uniform learning rates (green, $p < 1e-5$, Levene test). (**E**) The variability of stimulus selectivity is correlated with population coupling in networks with diverse learning rates (blue, $r = 0.18$, $p=1e-5$, Spearman correlation), but not in networks with uniform learning rates (green, $p=0.4$, Spearman correlation). Lines show linear regression fit for all datapoints (shaded coloured area indicates 95% confidence interval) (**F**) Dependence of network properties on the amplitude of injected noise ($\sigma_{\mathrm{OU}}$). Stimulus selectivity decreases with increasing $\sigma_{\mathrm{OU}}$ for networks with both

*Figure 2 continued on next page*

*Figure 2 continued*

diverse and uniform learning rates (blue and green lines, respectively). The distribution of population coupling broadens with increasing noise for networks with diverse learning rates, but not for networks with uniform learning rates (blue and green dashed lines, respectively). Panel A-E use $\sigma_{\mathrm{OU}} = 5.0$ (shaded gray area).

The online version of this article includes the following figure supplement(s) for figure 2:

**Figure supplement 1.** Dependence of the distribution of population coupling on the range of learning rate (**A-C**) and the amplitude of injected noise (**D**).

(*Okun et al., 2015*). This suggests that networks with diverse learning rates better match experimental observations than networks with uniform learning rates.

We next investigate the long-term variability of stimulus selectivity within both networks by measuring the fluctuations of neuronal stimulus selectivity throughout a period of synaptic plasticity (Materials and methods). We find that the magnitude of these fluctuations is independent of population coupling in the uniform network (p=0.4, Spearman correlation), but is correlated with population coupling in the diverse network (r = 0.18, p=1e-5, Spearman correlation *Figure 2E*).

We then characterise the dependence of population coupling and stimulus selectivity on the amplitude of external input noise, again for networks with either uniform or diverse $\alpha$ (*Figure 2F*). As the majority of excitatory synaptic input received by neurons in visual cortex is recurrent, we simulate a regime with relatively weak feedforward stimulus-related input and high noise for *Figure 2A-E* (*Cossell et al., 2015*; *Lin et al., 2015*). This results in a broader distribution of population coupling and weaker stimulus selectivity for networks with diverse $\alpha$, compared to networks with uniform $\alpha$ (*Figure 2F*). The dynamics of cortical activity observed in vivo are therefore more closely captured by networks with diverse $\alpha$, compared to networks with uniform $\alpha$.

Overall, these simulations show that the plasticity-coupling link observed in our small network model is robust in a larger network with receptive field properties and neuronal responses similar to mouse visual cortex. Networks with diverse $\alpha$ exhibit a broader range of population coupling than networks with uniform $\alpha$. Moreover, diverse learning rates introduce a correlation between a neuron's population coupling and its total excitatory synaptic input, in agreement with experimental observations (*Okun et al., 2015*). Taken together, diverse learning rates provide a parsimonious explanation for the diverse population coupling observed in sensory cortical networks.

## Experimental validation: population coupling is correlated with stimulus response variability in vivo

We have demonstrated that the population coupling of a neuron in a recurrent network model depends on its inherent plasticity. This plasticity-coupling link predicts a correlation between a neuron's population coupling and the variability of its stimulus selectivity. We now test this prediction using 2-photon calcium imaging of visual cortex in awake adult mice (Materials and methods). The data we analyse is publicly available and was collected by the Allen Institute for Brain Science (*Allen Brain Atlases and Data, 2016*). Mice passively viewed drifting or static gratings, interleaved with natural movies, while the simultaneous responses of $\sim 15,000$ excitatory neurons from 64 animals were recorded (*Figure 3A*). We measure the population coupling of each neuron over the entire recording session, and the preferred orientation of each neuron during the first 10 min and last 10 min of the experiment (Materials and methods, *Figure 3B*). We then compare these two measurements of orientation preference to identify whether the preferred orientation of some neurons vary over the course of the experiment.

There is a broad distribution of population coupling, in agreement with previous observations (*Figure 3C*; *Sedigh-Sarvestani et al., 2017*; *Okun et al., 2015*). Roughly 75% of neurons express variability of their preferred orientation or direction between the beginning and the end of the experiment. The distribution of changes in preferred orientation ($\Delta\mathrm{ORI}_{\mathrm{pref}}$) and preferred direction ($\Delta\mathrm{DIR}_{\mathrm{pref}}$) is skewed towards smaller magnitudes (*Figure 3D*).

Population coupling is weakly but significantly correlated with the average change in preferred orientation and in preferred direction (*Figure 3E*, r = 0.1, p<1e-5 and r = 0.12, p<1e-5, Spearman correlation). We characterise this dependence for each experiment by comparing the population coupling of neurons with variable preferences (those with $\Delta\mathrm{ORI}_{\mathrm{pref}} > 0$ or $\Delta\mathrm{DIR}_{\mathrm{pref}} > 0$) versus those

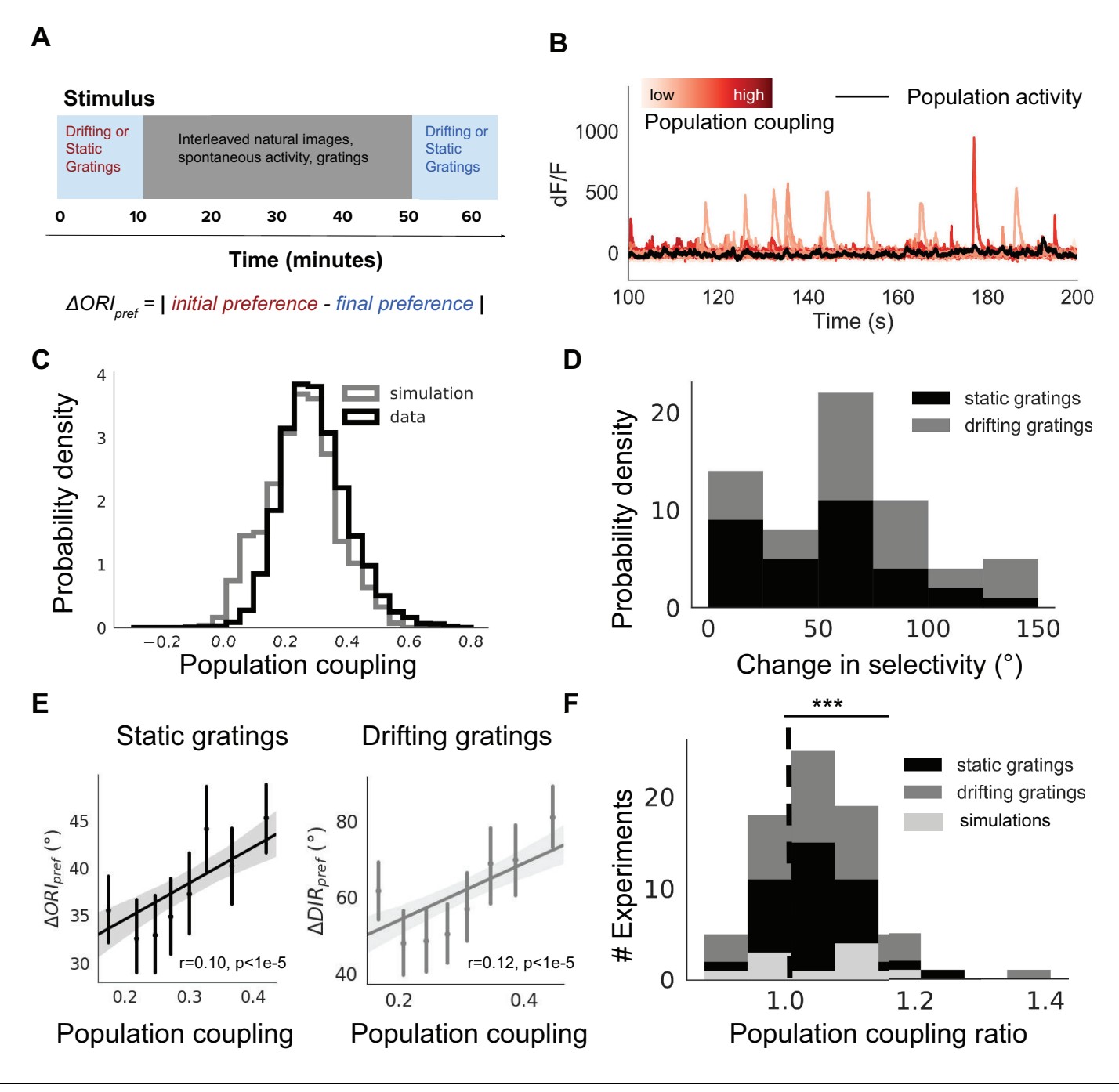

**Figure 3.** Population coupling is correlated with stimulus response variability in mouse visual cortex in vivo. (A) Diagram of stimulus and data analysis protocol (B) dF/F calcium fluorescence traces of neurons (red traces) in an example experiment from the Allen Brain Observatory. Mean activity is shown in black, and the population coupling of each neuron is indicated by its colour changing from light to dark red with increased population coupling. (C) The population coupling distribution of all neurons across all experiments (black, 64 experiments, 15,281 neurons) and in simulations (gray, 15 experiments, 3750 neurons). (D) Absolute difference in preferred orientation (black, static gratings, $\Delta\mathrm{ORI}_{pref}$) and preferred direction (gray, drifting gratings, $\Delta\mathrm{DIR}_{pref}$) between the beginning and the end of the recording session. (E) $\Delta\mathrm{ORI}_{pref}$ (left, black, static gratings) and $\Delta\mathrm{DIR}_{pref}$ (right, gray, drifting gratings) are correlated with population coupling. Data shown for all neurons with reliable stimulus responses across all experiments, binned by population coupling. Error bars for each bin show 95% confidence interval. Linear regression fit for all datapoints (shaded gray area indicates 95% confidence interval). (F) Distribution of ratios of the mean population coupling of neurons that change their preferred orientation ($\Delta\mathrm{ORI}_{pref} > 0$) or preferred direction ($\Delta\mathrm{DIR}_{pref} > 0$) versus mean population coupling of neurons that conserve their preferred orientation ($\Delta\mathrm{ORI}_{pref} = 0$) or preferred directions ($\Delta\mathrm{DIR}_{pref} = 0$), for each individual static grating experiment (black), drifting grating (gray) experiment, or network simulation (light gray).

*Figure 3 continued on next page*

*Figure 3 continued*

Dashed vertical line indicates expected value if a neuron's orientation or direction preference variability is not dependent on its population coupling (*** p<0.001, one sample t-test).

The online version of this article includes the following figure supplement(s) for figure 3:

**Figure supplement 1.** Orientation selectivity index is anti-correlated with population coupling.

**Figure supplement 2.** Dependence of plasticity-coupling link on response reliability in ABI experimental data.

**Figure supplement 3.** Changes in spatial frequency selectivity ($\Delta\mathrm{SF}_{\mathrm{pref}}$, (**A**)) and temporal frequency selectivity ($\Delta\mathrm{TF}_{\mathrm{pref}}$, (**B**)) are correlated with population coupling.

**Figure supplement 4.** Potentially confounding factor.

with stable preference. While there is substantial variability of the strength of the effect, the majority of experiments show a trend in which neurons with plastic orientation preferences have a higher mean population coupling than those with stable orientation or direction preferences (*Figure 3F*, p<0.001 for both static and drifting gratings, t-test). We observed similar correlations between population coupling and the average change in preferred spatial frequency and in preferred temporal frequency (*Figure 3—figure supplement 3E*, r = 0.054, p=0.001 and r = 0.058, p=0.001, Spearman correlation).

As the mean activity level of a neuron could conceivably determine its stimulus preference stability (*Ranson, 2017*; *Grosmark and Buzsáki, 2016*), we tested this and found no dependence of the tendency of a individual neuron to change stimulus preference on its average calcium fluorescence (p=0.88, Spearman correlation, *Figure 3—figure supplement 4*). We also found that this relationship between the change in selectivity and population coupling remained when controlling for peak calcium fluorescence or selectivity index as a potential confounding factor (*Figure 3—figure supplement 4*).

*Okun et al. (2015)* did not observe any correlations between population coupling of a neuron and its orientation selectivity. In contrast, our network model predicts that neurons with high population coupling are less selective than neurons with low population coupling. We tested whether there was this predicted dependence between population coupling and orientation selectivity in these data. We indeed found a weak anti-correlation between population coupling and orientation selectivity index (*Figure 3—figure supplement 1*, r = −0.05, p<1e-6, Pearson correlation).

## Diverse learning rates maintain both a stable backbone and a flexible substrate of stimulus representation

Our analysis thus far explored the impact of diverse rates of plasticity on synaptic connectivity. We established a link between diverse population coupling and diverse stimulus response variability, both of which are observed in sensory cortex. We now explore the functional implications of diverse population coupling and learning rates within recurrent networks. In order to simplify our analysis we consider both forms of diversity in isolation.

The presence of diverse rates of plasticity in a network suggests a dichotomy of roles: less plastic neurons could form stable stimulus representations while more plastic neurons could allow flexible representation. This could, for example, be beneficial during perceptual learning. We test this hypothesis by simulating an extended period of perceptual learning in our small network model (Materials and methods, *Figure 4A*). We do this using a simple paradigm in which a randomly chosen feedforward stimuli is associated with an increased external input. This external input could be mediated by a reward, or some other top-down signal. Hebbian plasticity potentiates the recurrent synaptic connections from neurons which are tuned to the stimulus onto all neurons. This increases the selectivity of all neurons to the associated stimulus (*Figure 4A*).

We evaluate the ability of our network to continually learn these stimulus associations in the case where $\alpha$ is slow for all neurons, $\alpha$ is fast for all neurons, or where there is diverse $\alpha$ (both slow and fast) for each feedforward stimulus group.

We find that a network with only fast $\alpha$ quickly learns the stimulus associations (*Figure 4A*, bottom). However, repeated associations with neurons that do not share feedforward stimuli cause the specificity of recurrent connectivity to decrease, thus degrading the representation of feedforward stimuli (*Figure 4A*, top). Although neurons still form associations with the feedforward stimulus, this

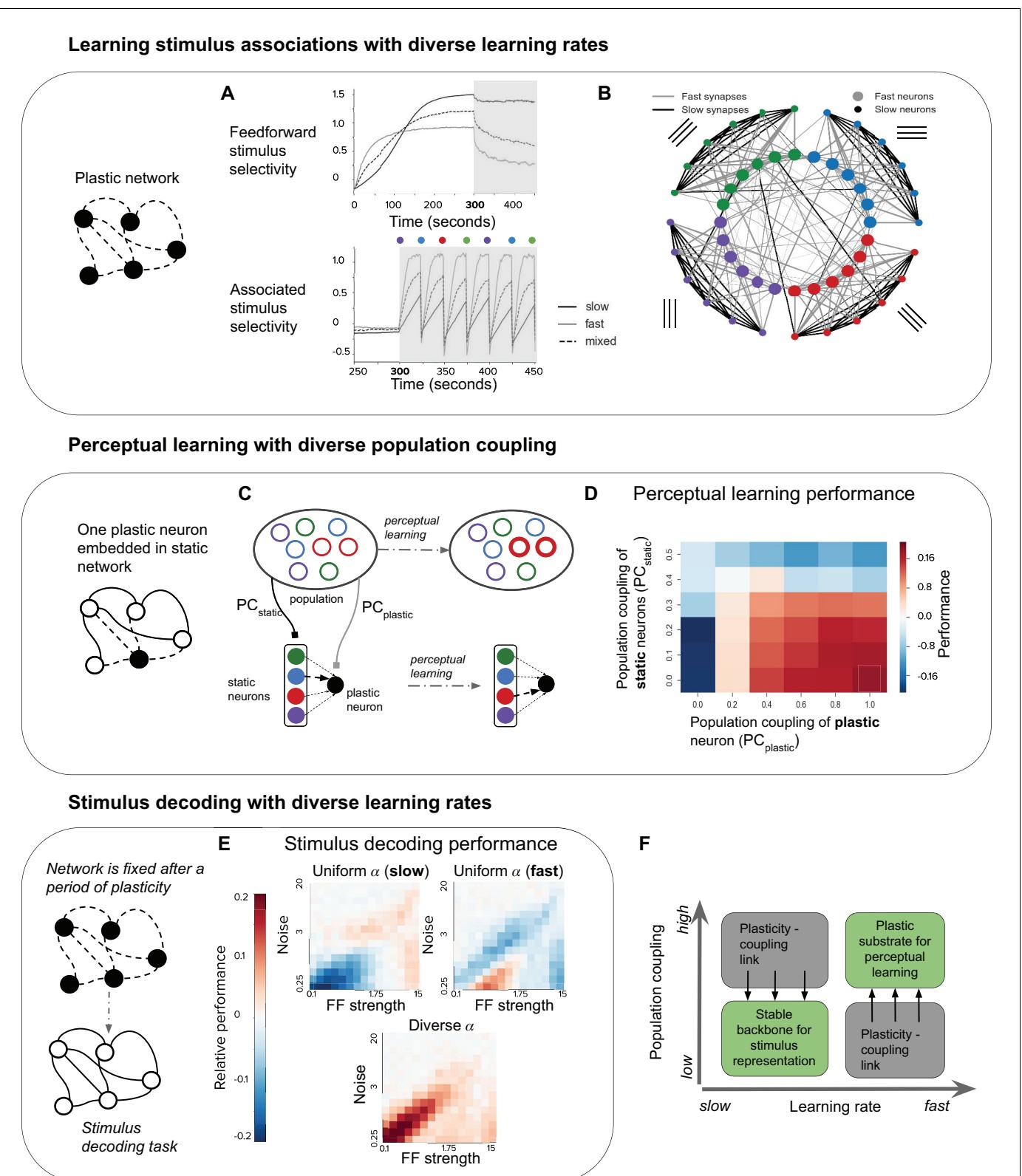

**Figure 4.** Plasticity-coupling link enables both robust stimulus representation and a flexible substrate for perceptual learning. (**A**) The evolution of mean selectivity to the feedforward stimulus (top) and a stimulus associated with an additional external input (bottom) in networks composed either entirely of neurons with slow $\alpha$ (black), fast $\alpha$ (gray), or a mix of both fast and slow $\alpha$ (dashed black). Shaded gray region indicates when the additional external input is present, and the coloured circles indicate the stimulus the external input is associated with at that time (this switches every 25 s) (**B**) Synaptic

*Figure 4 continued on next page*

*Figure 4 continued*

connectivity after plasticity for a network of neurons with slow (small circles) or fast (large circles) learning rates. Neurons in the network receive input selective to 1 of 4 possible stimuli (colour denotes stimulus preference). Synaptic inputs onto fast neurons and slow neurons are coloured gray and black respectively. The spatial organisation of neurons is for visualisation purposes only. (C–D) Investigating the impact of population coupling on perceptual learning. (C) Coupling of either the plastic neuron or static neurons to the population is set by adjusting $\mathrm{PC}_{\mathrm{plastic}}$ and $\mathrm{PC}_{\mathrm{plastic}}$ respectively. Perceptual learning is simulated through an additional external input whenever the preferred stimulus of the red neurons is present. This leads to the predominant synaptic weight onto the plastic neuron (black) switching from the neuron with the same original preferred stimulus (blue) to the neuron with the associated preferred stimulus (red). (D) Amount of perceptual learning which occurs at the plastic neuron, as the population coupling of either the plastic neuron ($\mathrm{PC}_{\mathrm{plastic}}$, x-axis) or static ($\mathrm{PC}_{\mathrm{static}}$, y-axis) is varied. Perceptual learning is quantified by the ratio of the red synaptic weight (associated stimulus) to the blue synaptic weight (original preferred stimulus of the plastic neuron) after plasticity. Red regions ($\Phi$) indicate successful perceptual learning, and occur only when $\mathrm{PC}_{\mathrm{plastic}}$ is high and $\mathrm{PC}_{\mathrm{static}}$ is low. (E) Relative stimulus decoding performance of fixed recurrent networks after a period of plasticity in order to develop the network. Networks were developed using either entirely neurons with fast learning rates, slow learning rates, or a 50/50 mix of both learning rates. The feedforward stimulus strength (x-axes) and noise (y-axes) were varied along a logarithmic scale. (F) Illustration of the synergistic effect of the plasticity-coupling link on perceptual learning. The plasticity-coupling link ensures that slow neurons have low population coupling and fast neurons have high population coupling, which panel D demonstrates is necessary for perceptual learning.

is because we keep the feedforward stimulus weights fixed; one can imagine that this feedforward selectivity may also degrade if these weights were plastic. Conversely, the network with only slow $\alpha$ retains a stable representation of the feedforward stimuli but performs poorly in representing the associated stimulus (*Figure 4A*). The network with diverse $\alpha$ overcomes these issues by having fast neurons which flexibly learn stimulus associations and slow neurons which maintain a 'backbone' of stimulus representation (see diagram, *Figure 4B*).

## The plasticity-coupling link enables efficient perceptual learning

Having demonstrated the advantage of diverse learning rates within a network for perceptual learning, we now ask whether diverse population coupling has any impact on a network's performance in this task. Given the plasticity-coupling link, we are particularly interested in whether the impact of population coupling on performance is dependent on a neuron's rate of plasticity. To investigate this, we choose the extreme case in which there is a single neuron with plastic synaptic inputs embedded in an otherwise static recurrent network. Since all other synapses in the network are static (see diagram, *Figure 4C*), we focus on how synaptic inputs onto the plastic neuron evolve during learning. We adjust the population coupling (PC) of either the single plastic neuron ($\mathrm{PC}_{\mathrm{plastic}}$) or the static neurons ($\mathrm{PC}_{\mathrm{static}}$), and measure the ability of the plastic neuron to learn a stimulus association (Materials and methods). We simulate perceptual learning by turning on an extra external input to all neurons in the network whenever the associated stimulus (red) is presented to the network (*Figure 4C*). We judge the plastic neuron to have learned the association if the synaptic weight from the presynaptic neuron selective to the associated stimulus becomes stronger than the weight from the presynaptic neuron selective to the non-associated stimulus (blue). We find that strongly coupling the plastic neuron to the population improves performance, while strongly coupling the static neurons to the population impairs performance (*Figure 4D*).

We can understand this by considering that, for learning to occur, synaptic potentiation must happen between the static neuron corresponding to the associated stimulus (red) and the plastic neuron. Increasing the plastic neuron's coupling to the rest of the population amplifies the correlation between the pre- and post- synaptic neuron when the associated stimulus is present, since the entire population receives an extra external input. On the other hand, strong coupling amongst the presynaptic static neurons decreases their stimulus selectivity, since they will be more co-active regardless of the stimulus identity. This corrupts the signal during stimulus association. These two effects combine, such that the new stimulus association is learned only when there is low population coupling amongst static neurons ($\mathrm{PC}_{\mathrm{static}}$) and high population coupling for the plastic neuron ($\mathrm{PC}_{\mathrm{plastic}}$) (*Figure 4D*, labelled $\Phi$). In order to enhance perceptual learning with diverse learning rates, plastic neurons should therefore be more coupled to the rest of the population than stable neurons. Correlated diversity of population coupling and plasticity helps achieve this (*Figure 2E*), ensuring that neurons best suited to the necessary stimulus representation remain stable, while neurons best suited to learning stimulus associations remain flexible. The plasticity-coupling link therefore efficiently

exploits the functional advantages conferred by both diverse learning rates and diverse population coupling.

## Diverse learning rates lead to networks with improved stimulus coding capabilities

Until now we have considered the effect of population coupling on a network's ability to learn stimulus associations. We are also interested in the impact of population coupling on a task that does not involve synaptic plasticity, since the differences in non-specific connectivity alone may affect a neuron's computational capability. We choose stimulus decoding as a simple example, and measure performance at decoding pairs of stimuli in a static network, after it has gone through a period of synaptic plasticity (Materials and methods). We compare three different network types; one which has been developed while it had only slow $\alpha$, one developed with only fast $\alpha$, and one developed with diverse $\alpha$ (*Figure 4E*). In a network with only slow $\alpha$, and therefore low population coupling, stimulus decoding performs relatively well when there are high levels of noise in the input. Networks with only fast $\alpha$ perform relatively well when there are low levels of noise. A network with diverse $\alpha$ seems to advantageously combine both of these properties, so that its performance is high across the entire range of input strength and noise levels.

## Discussion

We have studied the impact of diverse learning rates in a recurrent network model of visual cortex. Intriguingly, a plasticity-coupling link emerges in networks with diverse learning rates, in which neurons with fast learning rates are more coupled to population activity than neurons with slow learning rates. We substantiated a key prediction of our plasticity-coupling link with in vivo calcium imaging of mouse visual cortex from the Allen Brain Observatory (*Allen Brain Atlases and Data, 2016*), finding that a neuron is more likely to exhibit stimulus preference variability if it has high population coupling. Based on our findings we propose that the plasticity-coupling link efficiently combines stable and flexible stimulus representation.

### Stability and plasticity of stimulus responses

The architecture of a plastic substrate of neurons on top of a stable 'backbone' (*Figure 4B*) has been hypothesised before, and there is some compelling experimental evidence for this proposal (*Grosmark and Buzsáki, 2016*; *Clopath et al., 2017*; *Rose et al., 2016*; *Panas et al., 2015*). In particular, tracking of hippocampal cell assemblies reveal subsets of either plastic, highly active neurons or rigid, less active neurons (*Grosmark and Buzsáki, 2016*). Likewise, a statistical-mechanical analysis of network activity in hippocampal cell cultures identified both neurons which are highly active and contribute predominantly to network stability, and neurons which exhibit more long-term activity fluctuations without compromising overall network stability (*Panas et al., 2015*). In primary visual cortex - which we model - neurons exhibit characteristic fluctuations of their stimulus selectivity during baseline measurements, but nonetheless tend to retain their preferred stimulus following recovery from sensory deprivation (*Rose et al., 2016*). This provides evidence for a stable 'backbone' of recurrent connectivity which is resistant to sensory perturbations (*Clopath et al., 2017*). (*Ranson, 2017*) investigated the stability of locomotion-dependent modulation of visual responses across 14 days and, in contrast to *Grosmark and Buzsáki (2016)*, found that highly responsive neurons exhibited reasonably stable stimulus preference while weakly responsive neurons exhibit plastic stimulus preference. However, these experiments tracked different stimulus features - and over longer timescales - when compared with our study. Moreover, our inclusion of a homeostatic inhibitory plasticity rule that precisely controls excitatory firing rate precludes us from making predictions about the dependence of a neurons average firing rate and its propensity for stimulus preference plasticity (*Vogels et al., 2011*). Similar links between plasticity and population dynamics could emerge in other experiments that chronically image cortical network activity (*Driscoll et al., 2017*; *Singh et al., 2015*; *Peron et al., 2015*).

Since the majority of experiments which track stimulus preference evolution do so during visual discrimination paradigms, it is likely that top-down influences such as attention or reward modulation play significant roles in their observed dynamics (*Pakan et al., 2018*; *Caras and Sanes, 2017*; *Poort et al., 2015*; *Schoups et al., 2001*). An exception is (*Goltstein et al., 2013*), in which stimulus

preference is measured in the anaesthetised state, meaning that top-down inputs are likely to be absent. Likewise, (*Ranson, 2017*) tracked stimulus response stability during passive viewing, similar to the experimental setup of the data we analyse (*Allen Brain Atlases and Data, 2016*). As well as top-down modulation, further features missing from our network model are a realistic inhibitory circuitry (*Tremblay et al., 2016*; *Letzkus et al., 2015*), and incorporating changes in network dynamics which occur during sleep (*Grosmark and Buzsáki, 2016*; *Singh et al., 2015*), both of which are widely viewed to play an important role in regulating the plasticity of neural representation.

## A plasticity-coupling link in vivo

Our analysis of in vivo calcium imaging substantiates a key prediction of our network model by observing a correlation between the stimulus preference plasticity of a neuron and its population coupling (*Figure 3D*). Note that this relationship does not arise in our receptive-field network model with uniform learning rates (*Figure 2E*), so it is not a trivial consequence of any network model that exhibits diverse population coupling. Although the correlations we measured are quite small, this variability reflects what is observed in our network model (*Figure 2E*), and is not surprising given that there are likely many unobserved factors - aside from population coupling - which contribute to the dynamics of a neuron's observed stimulus response. Indeed, our network with uniform learning rates demonstrates significant stimulus response variability (*Figure 2E*, green), but crucially does not capture the correlation between this variability and population coupling which we observe both in vivo and in the network with diverse learning rates.

An advantage of the Allen Brain Observatory is the large amount of data and easily replicable data processing pipeline which allows us to build upon previous work investigating population coupling in the same dataset (*Sedigh-Sarvestani et al., 2017*). Since the population coupling of a neuron is correlated across brain states, and is only weakly dependent on stimulus type and mean fluorescence, we believe that it provides a good measure of a neurons functional integration within the local network, and would - according to our model - therefore provide a reasonable estimate of its propensity for perceptual learning (*Figure 4D*; *Sedigh-Sarvestani et al., 2017*; *Okun et al., 2015*). In agreement with our network model, and in contrast with observations from *Okun et al. (2015)*, population coupling is anti-correlated with orientation selectivity in the Allen Brain Observatory dataset (*Figure 3—figure supplement 1*). The disparity between these two experiments could be due to different experimental conditions, or the effect may not have been previously observed due to the smaller number of neurons used n = 431 in *Okun et al. (2015)*. Moreover, our observation that the changes in stimulus preferences ($\Delta\mathrm{ORI}_{\mathrm{pref}}$) are often non-zero but skewed towards small absolute values (*Figure 3C*) are in agreement with the hypothesis that stimulus preference is a slowly drifting property (*Rose et al., 2016*). Unfortunately, the experimental protocol limits us to directly comparing stimulus preference at only two timepoints; the beginning and end of a 62 min imaging session (*Figure 3A*). Nonetheless, significant changes in synaptic efficacies can be expressed within this time (*Meyer et al., 2014*). We hope that these findings will stimulate further experiments that allow us to more precisely test for the presence of a plasticity-coupling link across longer timepoints, and during learning.

While analysing fluctuations in stimulus selectivity in network simulations allow us to perform a like-for-like comparison with what could be measured from the experimental data, there are alternative approaches which may capture more complex stimulus representations (*Gallego et al., 2020*), and the plasticity of these representations. Exploring how plasticity relates to neurons involvement in such low-dimensional latent dynamics would be an interesting direction for further work.

## Population coupling and neuron function

Our network model provides a parsimonious explanation for the diverse population coupling recently observed in sensory cortex (*Okun et al., 2015*). Population coupling is dependent on the amount of recurrent synaptic input a neurons receives, in agreement with experimental data (*Figure 2C*). Note that this dependence is not present in networks with uniform learning rates, even though they too exhibit diverse population coupling. Moreover, the width of the population coupling distribution increases as the recurrent network approaches a dynamic regime dominated by high noise and diverse selectivity, typical in cortical networks (*Figure 2F*). These findings suggest that different population couplings may simply be a feature of varying learning rates and does not

necessarily mean (although we cannot exclude it) that the observed diversity reflects entirely different cell classes. Furthermore, one can imagine alternative mechanisms that lead to diverse population coupling in recurrent networks, such as imposing heterogeneous targets for the number of synaptic inputs received by each neuron. Investigating such alternative mechanisms was outside the scope of our study, but would provide an interesting avenue for further theoretical research.

The proposed plasticity-coupling link presents a counterintuitive interpretation of the role of 'soloists' and 'choristers' originally described by *Okun et al. (2015)*. While one may naively suppose that the weakly coupled 'soloists' are suited to undergo plasticity during learning, we propose that it is in fact the strongly coupled 'choristers' with a more plastic representation.

The functional impact of population coupling on learning is crucial: in order to enhance perceptual learning, plastic neurons in recurrent networks should be more coupled to the rest of the population than stable neurons (*Figure 4D,F*). We find that high population coupling helps plastic neuron change their stimulus preference towards an associated stimulus, but hinders the ability of stable neurons to provide an instructive signal for learning. Correlated diversity of population coupling and learning rate therefore enables both robust stimulus representation (low $\alpha$, PC) and a flexible substrate suitable for perceptual learning (high $\alpha$, PC). Strikingly, this relationship is precisely what the predicted plasticity-coupling link ensures (*Figure 4E*). Moreover, a recent theoretical study of sensory decoding proposed that untuned neurons contribute to decoding when they are correlated with tuned neurons (*Zylberberg, 2017*). Again, this is the relationship predicted by our model, since plastic neurons are less tuned than rigid neurons and are more strongly coupled to the population of (*Figure 1D,E*).

An alternative hypothesis for how population coupling relates to neural plasticity can be found in normative theories of learning. Consider a recurrent neural network trained with gradient descent to perform some function. Regardless of the function being learned, on average, neurons with greater impact on the rest of the network will have larger synaptic updates than other neurons. This is because the gradient of any function with respect to a neuron will depend on its ability to influence future activity, and neurons that contribute more to the gradient will be updated more. Neurons with a strong tendency to activate other neurons will have high levels of population coupling, while neurons with a tendency to inhibit other neurons (disynaptically, since these are excitatory neurons) will have low levels of population coupling. Neurons with limited impact will have moderate levels of population coupling. Thus, a gradient-based account would predict that there should be a U-shaped curve relating population coupling to plasticity. Our data showed may show some initial decrease in plasticity as population coupling increases (*Figure 3F*), but these questions outstrip the focus of this work. Consideration of whether alternative models based on gradients could explain this data may nonetheless be a fruitful avenue of enquiry for future experiments.

## Previous theoretical work

There are many previous theoretical explorations of how diversity in the synaptic plasticity of individual neurons affects learning. A recent study proposes a conceptually similar mechanism for modulating the stability or flexibility of memory formation, by implementing either symmetric or asymmetric STDP learning rules (*Park et al., 2017*). Diversity in synaptic learning rates was also explored within the traditional machine learning framework, whereby fast weights store temporary memories of recent events, compared with slow weights which capture regularities in input structure (*Ba et al., 2016*). Our work is related to previous approaches for overcoming catastrophic forgetting, which is often observed in neural networks during learning (*Grossberg, 1987*; *Carpenter and Grossberg, 1987*; *McClelland et al., 1995*; *Fusi et al., 2005*; *Roxin and Fusi, 2013*; *Benna and Fusi, 2016*). These approaches typically involve partitioning memories across timescales by implementing either synaptic states with different timescales, or neural architectures with different timescales. Here, we intead based our approach on experimental observations that suggest diverse learning rates within a sensory cortical network (*Ranson, 2017*; *Clopath et al., 2017*; *Poort et al., 2015*; *Lütcke et al., 2013*; *Rule et al., 2019*; *Rose et al., 2016*). Finally, individual synaptic updates in our model are defined by the learning rate of the postsynaptic neuron (*Equation 3*). Further work could explore whether our observed outcomes change if updates are instead dependent on the learning rate of the presynaptic neuron.

The plasticity-coupling link's impact on perceptual learning suggests a dichotomy of roles amongst neurons in a network, tied to a particular functional architecture: a stable 'backbone' of

stimulus representation formed by neurons with slow synaptic plasticity and low population coupling, on top of which lies a flexible substrate of neurons with fast synaptic plasticity and high population coupling. Diverse learning rates naturally enable this architecture, and offer a compelling candidate mechanism for mediating both forms of diversity - population coupling and stimulus response stability - recently observed in cortical networks. Finally, the plasticity-coupling link provides neuroscientists with a means to assess the tendency of particular neurons to influence future learning: those which are highly coupled to population activity are most likely to express plasticity. Ongoing advances in chronic multi-neuron calcium imaging, alongside neuron-specific optogenetic stimulation, will allow us to further probe and exploit these possibilities.

## Materials and methods

Our network model simulations were written in python 2.7 with numpy and scipy.

### Neuron model

For both the fully connected and the receptive field networks we use a simple firing rate neuron model, given by the transfer function $g(x)$ defined below, and as used previously by *Rajan et al. (2010)*; *Hennequin et al. (2014)*.

$$
\begin{aligned}
g(x) \quad &= 0 \text{ if } x{<}0 \\
&= (r_{\max} - r_0)\tanh(x/(r_{\max} - r_0)) \text{ if } x \geq 0.
\end{aligned}
\tag{1}
$$

This leads to firing rates with a baseline of $r_0$ and a maximum of $r_{\max}$. Following (*Rajan et al., 2010*), the firing rates $y_i$ of neuron $i$ driven by external input $H_i$ in a network are described below.

$$
\frac{dy_i}{dt} = -y_i + \sum_{j=1}^{N} W_{ij}g(y_j) + H_i,
\tag{2}
$$

where $W_{ij}$ is the weight of the synaptic connection from neuron $j$ to neuron $i$. All parameters are shown in *Table 1*.

### Modelling synaptic plasticity with diverse learning rates

We use a simple Hebbian learning rule with homeostatic synaptic scaling to model synaptic plasticity of recurrent excitatory to excitatory (E-E) synapses (*Gerstner and Kistler, 2002*),

$$
\frac{dW_{ij}^{EE}}{dt} = \alpha_i y_i y_j - \zeta \left( \sum_{k=1}^{N_E} W_{ik}^{EE} - W_{\text{total}}^{EE} \right)
\tag{3}
$$

where $\alpha_i$ is the learning rate of the postsynaptic neuron and $y_j$ and $y_i$ are the activities of the pre- and postsynaptic neuron respectively. $\zeta$ is the time constant of synaptic scaling, and $W_{\text{total}}^{EE}$ is the target amount of total recurrent synaptic input which each neuron can receive under the synaptic scaling rule.

This form of excitatory plasticity introduces competition amongst presynaptic synaptic weights and leads to the development of stimulus selectivity, as discussed in *Ko et al. (2013)*; *Clopath et al. (2010)*. We use a homeostatic rule to model inhibitory synaptic plasticity of recurrent inhibitory to excitatory (I-E) weights (*Vogels et al., 2011*),

**Table 1.** Simulation Parameters.

| $H_{\text{stim}}$ | 8.0 | $r_0$ | 1.0 | $r_{\max}$ | 20.0 | $y_0$ | 5.0 |
|---|---|---|---|---|---|---|---|
| $\alpha_{\text{s}}$ | $2.0{\times}10^{-6}$ Hz | $\zeta$ | $2.0{\times}10^{-4}$ Hz | $\eta$ | $1.0{\times}10^{-5}$ Hz | | |
| $w_{\max}$ | 0.042 | $w_{\max-\text{inh}}$ | 50 | $W_{\text{total}}^{EE}$ | 0.75 | | |
| $\sigma_{\text{OU}}$ | 1 | $\tau_{\text{OU}}$ | 10 ms | $W_{\text{init}}^{EE}$ | $0.5w_{\max}$ | $W_{\text{init}}^{IE}$ | 0.2 |

$$\frac{dW_{ij}^{IE}}{dt} = \eta y_j(y_i - y_0),\tag{4}$$

where $y_0$ is the homeostatic target firing rate, $\eta$ is the learning rate, and $W_{ij}^{IE}$ is the weight of the synaptic connection from inhibitory neuron $j$ to excitatory neuron $i$.

Excitatory weights are bounded so that their values lie between 0 and $w_{\max}$, and inhibitory weights are bounded so that they lie between $-w_{\max-\text{inh}}$ and 0.

While including two homeostatic mechanisms in our network model may seem redundant, they play different regulatory roles. Inhibitory plasticity largely controls the balance of excitation and inhibition received by a neuron, ensuring that it operates within its dynamic range. Synaptic scaling ensures that the total amount of recurrent excitation in the network is kept fixed as we vary its external input, while also introducing competition between presynaptic weights so that stimulus selectivity emerges. The synergistic effect of including multiple forms of plasticity has been widely studied in theoretical studies (*Zenke et al., 2015*; *Litwin-Kumar and Doiron, 2014*; *Triesch, 2007*; *Clopath et al., 2016*).

Note that the speed of all learning rates $\alpha$, $\zeta$, and $\eta$ are artificially increased in order to reduce the computational times resources required to simulate our network model. The timescales of synaptic plasticity in our network models are in the order of hundreds of seconds, while synaptic plasticity during perceptual learning occurs over the course of days in vivo. This increased learning rate does not qualitatively affect our results, as there is a sufficient separation of timescales between synaptic plasticity and network dynamics, and is a standard assumption in network models of synaptic plasticity (*Zenke et al., 2015*; *Litwin-Kumar and Doiron, 2014*).

## Fully connected network model

The fully connected network consists of $N_E$ excitatory neurons and a global inhibitory neuron ($N_I = 1$). The dynamics of both inhibitory (I) and excitatory (E) neurons are described by *Equation 1* and *Equation 2*. There is dense all-to-all synaptic connectivity in the E-E, E-I and I-E populations, and no I-I connectivity. Self-connections, or autapses, are not permitted in this network. $W$ in *Equation 2* is a square matrix with $(N_E + N_I)^2$ elements.

For *Figure 1*, we use a network with 48 excitatory neurons, and four input stimuli. Each neuron $i$ has a preferred stimulus $\theta_i^{\text{pref}}$, such that there are 12 neurons corresponding to each input stimulus. Each neuron receives its preferred stimulus input $H_{\text{stim}}$, and an independent noise source generated by an Ornstein-Uhlenbeck process, OU, with a mean of 0, variance of $\sigma_{\text{OU}}$ and correlation time $\tau_{\text{OU}}$. The external input $H_i$ to a neuron $i$ is therefore given by

$$H_i(t) = \delta(\theta_i^{\text{pref}} - \theta_{\text{input}}(t))H_{\text{stim}} + \text{OU}_i(t),\tag{5}$$

where $\delta$ is the Kronecker delta function. For *Figure 1B–D*, these input groups are further divided so that there are six slow neurons (with $\alpha_i = \alpha_s$) and six fast neurons (with $\alpha_i = 5\alpha_s$) per group. For *Figure 1E*, each input group of 12 neurons contains a single neuron corresponding to each of the 12 learning rates. The learning rates are logarithmically spaced between $0.5\alpha_s$ and $75\alpha_s$.

All excitatory-to-inhibitory synapses are uniformly initialised with weights $W_{\text{init}}^{\text{EE}}$, excitatory-to-inhibitory synapses with weights $W_{\text{init}}^{\text{EI}}$, and inhibitory-to-excitatory synapses with weights $W_{\text{init}}^{\text{IE}}$. We simulate the evolution of synaptic weights during visually evoked activity by sequentially presenting the network with a randomly chosen stimulus from the four input stimuli. Each stimulus is presented for 500 ms. The total simulation time is 500 s, and synaptic weights are updated at each timestep with the learning rules given by *Equation 3* and *Equation 4*.

For *Figure 1D*, connection specificity is defined as the average ratio of specific to non-specific excitatory synaptic weights received by neurons. Synaptic inputs from neurons in the same input stimulus group as the postsynaptic neuron are specific (i.e. they share the same feedforward stimulus preference), while all other synaptic inputs are non-specific. Specificity fluctuations are defined as the standard deviation of the connection specificity over time, where specificity is sampled every second from 200 to 500 seconds.

## Measuring population coupling

As introduced by *Okun et al. (2015)*, the population coupling $\mathrm{PC}_i$ of a neuron $i$ is measured by calculating the Pearson correlation coefficient of each neurons' activity $x_i$ with the average activity of the rest of the population;

$$\mathrm{PC}_i = \mathrm{corr}(x_i, \frac{1}{N-1}\sum_{j\neq i}^{N} x_j). \tag{6}$$

Synaptic weights are kept fixed during the population coupling measurement, while external input is as in *Equation 5*. We measured population coupling using 250 s of activity.

## Receptive-field based network model

For *Figure 2*, we adapt a previously developed model of receptive field properties in mouse visual cortex (*Watanabe et al., 2016*). We add neuronal dynamics and, beginning with uniform connectivity, simulate synaptic plasticity as visual stimuli are presented to the network. This model is constructed by assigning receptive fields to each excitatory neuron from a 2D Gabor function,

$$\begin{aligned}
\mathrm{RF}(x', y') &= \mathrm{A}\exp(\frac{-x'^2}{2\sigma_x^2} - \frac{-y'^2}{2\sigma_y^2})\cos(2\pi f x' + \phi) \\
x' &= x\cos\theta - y\sin\theta \\
y' &= x\sin\theta + y\cos\theta
\end{aligned} \tag{7}$$

where A is the amplitude, $\sigma_x$ and $\sigma_y$ are the standard deviations of the Gaussian, $\theta$ is the orientation, $f$ is the frequency and $\phi$ is the phase of the receptive field. A network of 250 excitatory neurons with receptive fields is randomly generated from *Equation 7*, with $f = 2$, $\sigma_x = \sigma_y = 0.5$, $\phi \sim (0, 2\pi)$, $\theta \sim \{\pi/4, \pi/2, 3\pi/4, ..., 2\pi\}$. As in the previous network, there is a single inhibitory neuron which all excitatory neurons project to, and receive inhibition from.

Neurons are rate-based and have similar dynamics as in the simple network model (*Equation 1*, *Equation 2*). Synaptic plasticity is also governed by the same learning rules (*Equation 3*, *Equation 4*). Inputs are presented to the network in the form of 2D images, and the input to each neuron $i$ for a given image $I_{\mathrm{ext}}$ is determined by the pixel-wise dot product of that image with the neurons' receptive field $\mathrm{RF}_i$, in addition to an independent noise term for each neuron given by an Ornstein-Uhlenbeck process;

$$H_i(t) = I_{\mathrm{ext}} \cdot \mathrm{RF}_i + \mathrm{OU}_i(t). \tag{8}$$

All excitatory-to-inhibitory synapses are uniformly initialised with weights $W_{\mathrm{init-RF}}^{\mathrm{EE}}$ and inhibitory-to-excitatory synapses with weights $W_{\mathrm{init-RF}}^{\mathrm{IE}}$. We simulate the evolution of synaptic weights during visually-evoked activity by sequentially presenting the network with randomly chosen bars of different orientations. Each image is presented for 500 ms. The total simulation time is 500 s, and synaptic weights are updated at each timestep. All results in *Figure 2* are pooled from 15 independent network instances, with 250 excitatory neurons in each network instance.

We define the selectivity of each neuron as $\bar{w}_{\mathrm{specific}} - \bar{w}_{\mathrm{non-specific}}$, where $w_{\mathrm{specific}}$ are the synaptic weights from neurons which share the same receptive field orientation and $w_{\mathrm{non-specific}}$ are the synaptic weights from neurons which have a different receptive field orientation.

## The allen brain observatory: 2-photon calcium imaging of visual responses in vivo

We use data from the Allen Brain Observatory, a publicly available and curated survey of neural activity in adult mouse visual cortex. A comprehensive description of the experimental methods, data acquisition and data analyses are available as white papers published by the Allen Institute for Brain Science (*Allen Brain Atlases and Data, 2016*). A list of the experiment IDs used in our analysis is available at 10.6084/m9.figshare.11837406.

Briefly, GCaMP6F was expressed in primary visual area neurons of transgenic mice line Ai93. Cranial surgery was performed to insert a window between p37-p63, followed by 2 weeks of habituation to the experiment setting. Mice were head-fixed on top of a rotating disk and could walk freely. 2-photon imaging experiments were conducted as the mouse passively viewed the stimulus protocol

on a screen. The stimulus protocols included in our analysis consisted of either i) 10 min of drifting gratings, followed by 42 min of interleaved natural movies, drifting gratings and spontaneous activity, followed by another 10 min of drifting gratings, or ii) 8 min of static gratings, followed by 45 min of interleaved natural movies, static gratings and spontaneous activity, followed by 9 min of static gratings (*Figure 3A*, see white paper for further details). Drifting gratings were presented at eight uniformly separated directions and at five different temporal frequencies. Static gratings were presented at six uniformly separated orientations separated, five different spatial frequencies and four different phases. 112 imaging experiments were initially included in our analysis.

## Measuring population coupling and stimulus response variability in vivo

The Allen Brain Atlas API provides functions which allow us to extract fluorescence traces and measure average stimulus response properties of individual cells during an experiment (*Allen Brain Atlases and Data, 2016*). Motion correction, ROI detection and segmentation, and the removal of neuropil fluorescence artefacts are automatically performed using the API. We customised scripts within this API so that we could measure population coupling and stimulus response properties under specific conditions and timeframes.

We measure population coupling similarly to the network model analysis (*Equation 6*), but where a neuron's activity is represented by calcium fluorescence dF/F. To ensure a reliable estimate of the population activity when calculating population coupling, we exclude any experiments in which less than 50 neurons were recorded. We also exclude any experiments in which the population couplings are not sufficiently consistent when using either half (randomly chosen) of the neurons to estimate population activity ($r^2<0.8$, linear regression). This reduces the number of experiments in our analysis from 112 to 64, for a total of 15,281 neurons. In addition, we only include neurons with a response reliability that is above the median reliability. The response reliability is defined as the trial-to-trial correlation of the dF/F traces during the neuron's preferred stimulus presentation (*Allen Brain Atlases and Data, 2016*) . *Figure 3—figure supplement 2* shows a similar analysis, but where we include neurons with the top 75% most reliable neurons, or include all neurons, regardless of their response reliability.

We measure the preferred orientation of each neuron during both the first presentation of static or drifting gratings ($\mathrm{ORI}_{\mathrm{pref}-1}$), and the final presentation of static or drifting gratings ($\mathrm{ORI}_{\mathrm{pref}-2}$) (*Figure 3A*). The preferred orientation is defined as the grating that evoked the largest mean response across all trials. Note that each experiment contains only either static or drifting gratings, so there is no overlap between these two conditions. The absolute difference in preferred orientation is calculated as: $\Delta\mathrm{ORI}_{\mathrm{pref}} = |\mathrm{ORI}_{\mathrm{pref}-1} - \mathrm{ORI}_{\mathrm{pref}-2}|$. Similarly, the absolute difference in preferred direction is calculated as: $\Delta\mathrm{DIR}_{\mathrm{pref}} = |\mathrm{DIR}_{\mathrm{pref}-1} - \mathrm{DIR}_{\mathrm{pref}-2}|$.

## Simulating perceptual learning

For the perceptual learning simulation in *Figure 4A*, the input $H_i$ to each neuron is simulated as before, but with an additional term which is active whenever the stimulus associated with the additional external input is present (*Equation 9*). We first simulate synaptic plasticity without any stimulus associations for 300 s (i.e. with $H_{\mathrm{associated}} = 0$), and then simulate perceptual learning (with $H_{\mathrm{associated}} = 10$). The identity of the associated stimulus is changed every 25 s to simulate continual learning.

$$H_i(t) = \delta(\theta_i^{\mathrm{pref}} - \theta_{\mathrm{input}}(t))H_{\mathrm{stim}} + \delta(\theta^{\mathrm{associated}} - \theta_{\mathrm{input}}(t))H_{\mathrm{associated}} + \mathrm{OU}_i(t), \tag{9}$$

where $\delta$ is the Kronecker delta function. Feedforward stimulus selectivity is defined as $\frac{\bar{w}_{\mathrm{specific}}}{\bar{w}_{\mathrm{non-specific}}} - 1$, where $w_{\mathrm{specific}}$ are the synaptic weights from neurons which share the same feedforward stimulus preference and $w_{\mathrm{non-specific}}$ are the synaptic weights from neurons which have a different feedforward stimulus preference. Likewise, associated stimulus selectivity is defined as $\frac{\bar{w}_{\mathrm{associated}}}{\bar{w}_{\mathrm{non-associated}}} - 1$, where $w_{\mathrm{associated}}$ are the synaptic weights from neurons whose feedforward stimulus preference is the associated stimulus, and $w_{\mathrm{non-associated}}$ are the synaptic weights from other neurons.

## Single plastic neuron embedded in a static network

In order to systematically investigate the effect of population coupling on perceptual learning, we must keep the population coupling of both the static and plastic population fixed throughout the experiment. Since changes in the synaptic weights connecting both these populations will alter their population coupling, we overcome this by making these particular synapses functionally silent. That is, while their synaptic weight is updated depending on pre- and post-synaptic activity as before (*Equation 3*), these synapses do not contribute when calculating the activity of the static and plastic neurons. The activities of the static and plastic neurons ($y^{\text{static}}$ and $y^{\text{plastic}}$) are therefore only determined by their external input and the activities of the population ($y^{\text{pop}}$, *Figure 4C*), meaning that their population coupling can be systematically varied:

$$\frac{dy_i^{\text{pop}}}{dt} = -y_i^{\text{pop}} + \sum_{j=i}^{N} W_{ij}g(y_j^{\text{pop}}) + H_i,$$
$$\frac{dy_i^{\text{plastic}}}{dt} = -y_i^{\text{plastic}} + \text{PC}_{\text{plastic}} \sum_{j=i}^{N} W_{ij}g(y_j^{\text{pop}}) + H_i, \tag{10}$$
$$\frac{dy_i^{\text{static}}}{dt} = -y_i^{\text{static}} + \text{PC}_{\text{static}} \sum_{j=i}^{N} W_{ij}g(y_j^{\text{pop}}) + H_i.$$

$W_{ij}$ is fixed for the duration of the simulation, while the synaptic weights from the static to the plastic population are updated as below;

$$\frac{dW_{ij}^{plastic}}{dt} = \alpha y_i^{\text{static}} y_j^{\text{plastic}} - \zeta \left( \sum_{k=1}^{N_{\text{plastic}}} W_{kj}^{plastic} - W_{\text{total}}^{EE} \right). \tag{11}$$

As before, the input $H_i$ has an additional term which is active whenever the stimulus associated with the additional external input is present (*Equation 9*, that is whenever the red stimulus is being presented). We first simulate synaptic plasticity without any stimulus associations for 500 s (i.e. with $H_{\text{associated}} = 0$), and then simulate perceptual learning (with $H_{\text{associated}} = 10$) for 100 s. Perceptual learning is quantified by the ratio of the red synaptic weight (associated stimulus) to the blue synaptic weight (original preferred stimulus of the plastic neuron) after plasticity.

## Measuring stimulus decoding performance

We train a perceptron to decode the stimulus identity from the individual activity of all neurons in the network, using the scikit-learn python package. The average activity of each neuron across a 500 ms sampling period are used as inputs during training. For *Figure 4E*, performance at decoding pairs of stimuli simultaneously presented to the network is shown. Relative deviation from the average performance of a perceptron trained to decode pairs of stimuli (28 possible pairs from eight stimuli) over all three network types is calculated. The relative deviations for each network type from the average across all networks types are shown (*Figure 4E*).

## Code availability

Code is publicly available on GitHub on https://github.com/yannaodh/sweeney_clopath_2020 (copy archived at https://github.com/elifesciences-publications/sweeney_clopath_2020; *Sweeney, 2020*).

## Acknowledgements

We are grateful to Tobias Rose and Alex Cayco-Gajic for their invaluable scientific feedback and comments on the manuscript. This work was funded by BBSRC BB/N013956/1, BB/N019008/1, Wellcome Trust 200790/Z/16/Z, Simons Foundation 564408 and EPSRC EP/R035806/1. We are also grateful to the Allen Institute for Brain Science for making freely available their data and tools.

## Additional information

### Funding

| Funder | Grant reference number | Author |
|---|---|---|
| Biotechnology and Biological Sciences Research Council | BB/N013956/1 | Claudia Clopath |
| Biotechnology and Biological Sciences Research Council | BB/N019008/1 | Claudia Clopath |
| Wellcome | 200790/Z/16/Z | Claudia Clopath |
| Engineering and Physical Sciences Research Council | EP/R035806/1 | Claudia Clopath |
| Simons Foundation | 564408 | Claudia Clopath |

The funders had no role in study design, data collection and interpretation, or the decision to submit the work for publication.

### Author contributions

Yann Sweeney, Conceptualization, Resources, Data curation, Software, Formal analysis, Validation, Investigation, Visualization, Methodology, Writing - original draft, Writing - review and editing; Claudia Clopath, Conceptualization, Formal analysis, Supervision, Funding acquisition, Validation, Investigation, Visualization, Methodology, Writing - original draft, Project administration, Writing - review and editing

### Author ORCIDs

Yann Sweeney (ID) https://orcid.org/0000-0002-2164-2438
Claudia Clopath (ID) https://orcid.org/0000-0003-4507-8648

### Decision letter and Author response

Decision letter https://doi.org/10.7554/eLife.56053.sa1
Author response https://doi.org/10.7554/eLife.56053.sa2

## Additional files

### Supplementary files

• Transparent reporting form

### Data availability

All calcium imaging data came from the Allen Institute for Brain Science, Allen Brain Observatory. Available from: http://observatory.brain-map.org/visualcoding/. A list of experiment IDs can be found on FigShare, under the doi 10.6084/m9.figshare.11837406. Code is publicly available on GitHub on https://github.com/yannaodh/sweeney_clopath_2020 (copy archived at https://github.com/elifesciences-publications/sweeney_clopath_2020).

The following dataset was generated:

| Author(s) | Year | Dataset title | Dataset URL | Database and Identifier |
|---|---|---|---|---|
| Sweeney Y | 2020 | Supplementary table 1 (List of ABI experiment IDs) | https://figshare.com/articles/Supplementary_table_1_List_of_ABI_experiment_IDs_/11837406 | figshare, 10.6084/m9.figshare.11837406 |

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

## Appendix 1

### Supplementary materials

### Methods for Figure S1

We simulate a network of 1 postsynaptic neuron and 10 presynaptic neurons. 3 of the presynaptic neurons share the same stimulus preference as the postsynaptic neuron and the remaining have different preferred stimuli. We then identify the parameter regime in which the coupling of the single plastic neuron to the rest of the population is correlated to its learning rate. To do so, we measure the population coupling of the postsynaptic neuron for a range of different learning rates from $0.5\alpha_s$ to $10\alpha_s$, using a separate network instantiation for each value of $\alpha$. We then estimate the slope of the relationship between population coupling and $\alpha$ using linear regression, across a range of values for the synaptic scaling rate ($\zeta$) and noise magnitude ($\sigma_{OU}$).

This can be expressed by the terms in (**Equation 12**), below, which describe a synaptic weight update dW under Hebbian plasticity with learning rate $\alpha$ and synaptic scaling with rate $\zeta$. For specific weights, the correlation with the input, corr(stim), is high, meaning that these weights tend to be large. For non-specific weights, corr(stim) is close to 0 and so does not contribute to the update dW. From the terms in (**Equation 12**), we can see that a large $\alpha$, the presence of noise, and a small $\eta$, can all contribute to large non-specific weights.

$$dW = \alpha(\mathrm{corr}(\mathrm{stim}) + \mathrm{corr}(\mathrm{noise})) - \zeta(W_{\mathrm{total}} - W_{\mathrm{target}}) \tag{12}$$

