## [Decision Letter]

**Acceptance summary:**

Many brain models work with just the mean activity across a population, ignoring the wide diversity of response properties that is characteristic of many neuronal population recordings. The authors provide modelling results which point to an intriguing link between the diversities of response variability and synaptic learning rates. A signature of this diversity link is seen in a relation between how malleable a neuron's tuning is over time and how coupled a neuron's activity is to the whole population. This prediction was validated with population recordings in mouse V1, giving support for their modelling work. Building models that explore the inherent diversity of neuronal response, in its various forms, is an important avenue of study and this work provides a critical step in this direction.

**Decision letter after peer review:**

[Editors’ note: the authors submitted for reconsideration following the decision after peer review. What follows is the decision letter after the first round of review.]

Thank you for submitting your work entitled "Population coupling predicts the plasticity of stimulus responses in cortical circuits" for consideration by *eLife*. Your article has been reviewed by three peer reviewers, and the evaluation has been overseen by a Reviewing Editor and a Senior Editor. The following individuals involved in review of your submission have agreed to reveal their identity: Blake A Richards (Reviewer #3).

Our decision has been reached after consultation between the reviewers. Based on these discussions and the individual reviews below, we regret to inform you that your work will not be considered further for publication in *eLife*.

There was considerable enthusiasm for the approach and for the novel hypothesis that, in a network with diverse learning rates, neurons with fast rates are more coupled to population activity than neurons with slow rates. The attempt to substantiate this hypothesis by using recordings from the Allen Brain Observatory was considered a considerable strength. However as amply stated in the original reviews, the strength/shape/significance of the relationships in Figures 3E,F are not firmly established and some more compelling insight into the diversity of timescales in the actual data is definitely needed.

A new submission of a thoroughly reworked paper that addresses these concerns and additional concerns in original reviews could be viewed favorably at *eLife*. The Reviewing Editor would particularly refer the authors to concerns by reviewer #1 "I would have expected an associated Figure 3E and F from the model so that a better comparison could be made and we could understanding how much the α's need to vary." and "The relation between learning diversity and population coupling is interesting – but outside of the initial observation in the model there is no real theory/understanding for this relation being offered", and from reviewer #3 "They begin with diverse learning rates and then examine the effect on population coupling. But, what if, functionally, population coupling should determine learning rate, as stipulated by gradient descent?"

Reviewer #1:

The paper by Sweeny and Clopath investigates how a diversity of plasticity timescales allows for a diversity of neuronal coupling to population activity. Overall, the paper investigates an interesting effect. However, I feel that the paper falls short in several areas. I explain these below.

1) While the comparison to the Allen dataset is inspired, the end result is somewhat disappointing. The relation between change in preferred orientation and population coupling shown in Figure 3E and F are quite weak (I realize that it is statistically significant). Further, there is no attempt to relate the weak trend in the data to the distribution of learning timescales that would be required to produce it. This is likely because the model is simplified and there is not a real sense of continuous tuning (only four orientations were shown). In this way the model prediction is not really tight. I would have expected an associated Figure 3E and F from the model so that a better comparison could be made and we could understanding how much the α's need to vary.

2) The relation between learning diversity and population coupling is interesting – but outside of the initial observation in the model there is no real theory/understanding for this relation being offered. Is there a simple mean field style theory that can be put forth? Can other sources of heterogeneity (say single neuron baseline firing rate r_0_) create a similar relation? Why is there a spread of population coupling in the uniform α case? Answers to these questions could help build a deeper understanding of the key result. As it stands the paper simply shows simulation results and gives a loose connection between synaptic variability, population coupling, and response tuning. I expect more in a high profile publication.

Reviewer #2:

Sweeney and Clopath report simulations and analysis of in vivo calcium imaging data suggesting that neurons with faster learning rates will exhibit higher population coupling, consistent with the idea of functional network architecture composed of slowly-changing connectivity between stably tuned neurons and quickly changing connectivity between unstably tuned neurons.

The authors begin by studying networks where they train a low-dimensional structure into the excitatory weights (four-dimensional in Figure 1, and I would guess 8-dimensional in Figure 2 due to the 8 orientations of receptive field), with one global inhibitory neuron. Population coupling describes a one-dimensional correlation structure – is this an appropriate description for these network, or are the activity four or eight dimensional? How do the correlations within- and between the trained structures relate to the population coupling, and how does the timescale of a neuron's plasticity relate to its coupling to co-stimulated neurons vs non-co-stimulated neurons?

What is the relationship between the timescale of the associations trained in Figure 4, and do the results depend on the relationship between that timescale of stimulation and the fast/slow timescales of plasticity?

I applaud the authors for seeking a direct experimental test of their model's results. The results of that test (Figure 3E) seem, however, fairly weak. More importantly, it seems that the basic characterization of tuning stability in this data has been skipped over, which makes it difficult to interpret these results.

The trial-by-trial responses to gratings (at least in the Allen Brain Observatory dataset) exhibit significant variability, and the tuning curve is often a poor predictor of the responses. This requires, I think, a stronger analysis to test the authors' prediction.

a) First off, how stable is the tuning? Is orientation selectivity stable? If there are neurons that are tuned only in the early or the late epoch, how does that match with the model?

b)In Figure 3B, it seems that the neurons with the largest responses are the most population coupled, which makes sense given the definition of population coupling. How does this contribute to the neurons' orientation selectivity and relate to tuning stability?

c) In neurons that are significantly tuned at the early and late epochs, is their tuning different? This is close to what the authors asked, but I think it could be made more rigorous – one idea is by a statistical test asking whether the distribution of responses to the initially preferred stimulus is distinguishable between the early and late grating epochs. (And correspondingly for the preferred stimulus of the late epoch.)

d)In Figure 3B, it seems that the neurons with the largest responses are the most population coupled, which makes sense given the definition of population coupling. Does this hold up with extracted events? How does this contribute to the neurons' orientation selectivity and relate to tuning stability?

e) How do these results relate to the degree of neurons' tuning and the reliability of their responses? Do strongly tuned neurons have more stable stimulus preferences? Do neurons with lower trial-to-trial variability have more stable stimulus preferences?

f) The authors refer only to orientation tuning. For the drifting gratings, what about direction? (And what about spatial or temporal frequencies?)

g) The description of which data from the Allen Brain Observatory were used seems incomplete. How were experiments chosen to include; which Cre lines, layers, and areas were the data used from? Do the results differ across those dimensions? That dataset currently includes a lot more than the 112 experiments with gratings included.

h) The observatory API also included extracted events, to account for the autocorrelation time of the fluorescence. Do the results depend on the use of df/f compared to events?

Reviewer #3:

In this paper, Sweeney and Clopath examine the relationship between "population coupling" and synaptic plasticity, both in a series of computational models and experimental data from the Allen Brain Institute (2-photon calcium imaging in mouse primary visual cortex). The metric used for population coupling here is the same one introduced by Okun et al., 2015, namely, the extent to which a given neuron correlates with the average activity of the entire population.

In their first model, the authors examine the impact of diverse learning rates on excitatory synapses in a fully connected model of rate-based, leaky neurons. The excitatory synapses are updated using a Hebbian rule (with some competition for stabilization). The Hebbian updates are controlled with learning rates \α_i_, where i is the index of the neuron. Thus, each neuron can have its own learning rate. The authors demonstrate that neurons with fast learning rates tend to have less specificity in their connections, and more fluctuations in that specificity as well. Furthermore, they show that there is a relationship between learning rate and population coupling, with higher population coupling for neurons with larger learning rates. They also show that diverse population coupling is a natural consequence of diverse learning rates. Thus, their model predicts that if cortical neurons possess a diverse set of learning rates, then there should be a high degree of variability in population coupling, and neurons with greater population coupling should be those that are most plastic.

The authors then examine whether these predictions from their model are borne out in the Allen data. They demonstrate that there is indeed a high diversity in population coupling of excitatory neurons (as observed by Okun et al.). As well, they show that neurons whose preferred stimulus orientation changes most over the course of the recording session are those with the greatest degree of population coupling, as predicted by their model.

Finally, the authors examine the functional implications of diverse learning rates in networks with sensory inputs and potential for learned associations. They show that diverse learning rates in their model helps to maintain the ability of the neurons to carry information about stimulus identity while permitting continual learning of new stimulus associations.

Overall, I think this is an excellent paper. I really enjoyed it. The introduction of a computational model that makes clear experimental predictions, followed by an explicit test of those predictions in available large datasets is exactly how computational neuroscience should proceed, and I applaud the authors for this. I also appreciated the attempt to show the functional importance of diverse learning rates at the end of the paper.

I have two related concerns about the findings in the paper, though, which I would encourage the authors to consider. The root of my concerns is the lack of a normative framework for understanding learning in both the models and the data. The authors begin with diverse learning rates as a potential phenomenon, then attempt to make predictions and explore functional implications. I wonder whether flipping this might not have been a more fruitful approach for explaining the experimental data. That is, the authors could have started with a question of what functional problem is to be solved, then showed that diverse learning rates could solve it, and then confirmed the phenomenon in the data. There are two reasons why this more normative approach may be better at explaining the data:

1) An alternative account for the phenomenon the authors observe in the Allen data can be found in normative models. Consider a recurrent neural network trained with gradient descent to perform some function. Regardless of the function being learned, on average, neurons with greater impact on the rest of the network will have larger synaptic updates than other neurons. This is because the gradient of any function with respect to a neuron will depend on its ability to influence future activity, and neurons that contribute more to the gradient will be updated more. Now, neurons with a strong tendency to activate other neurons will have high levels of population coupling, while neurons with a tendency to inhibit other neurons (disynaptically, since these are excitatory neurons) will have low levels of population coding. Neurons with limited impact will have middle levels of population coupling. Thus, a gradient based account would predict that there should be a U-shaped curve relating population coupling to plasticity. To my eye, that is precisely what the Allen data actually shows (Figure 3E). Thus, I worry that the authors have it backwards. They begin with diverse learning rates and then examine the effect on population coupling. But, what if, functionally, population coupling should determine learning rate, as stipulated by gradient descent? Now, to be clear, I do not think the authors have to actually address this hypothesis I put forward with new simulations. Per the *eLife* philosophy, I would like to improve this paper, not demand the paper I may have been inclined to write myself. So, I don't think new simulations are needed. However, given the slightly questionable fit of their model hypothesis to the data shown in Figure 3E, I think that this alternative hypothesis should at least be mentioned in the Discussion.

2) The purpose of the simulations in Figure 4 were a bit mysterious to me. The authors appeared to make fairly strong assumptions about what would be useful to a network, e.g. that it would be good to show increased selectivity to some external stimulus, or that information should be as preserved as possible. But, why not start with the more obvious normative framework for multiple learning rates, and one which the authors identify in the Discussion, i.e. the need to avoid catastrophic forgetting? This is sort of demonstrated in Figure 4, but in an indirect fashion that relies on the assumption that selectivity is a good performance metric. Why not actually use the networks to perform various tasks, and show that the networks are better at avoiding catastrophic forgetting of the tasks when diverse learning rates are present? Even a simple task would be easier to understand the implications, than the selectivity measurements provided by the authors, in my opinion. But, again, I don't actually think new simulations would be required, per se (though they could help). Nonetheless, I think a more intuitive account for why selectivity is the right metric to use would be helpful.

[Editors’ note: further revisions were suggested prior to acceptance, as described below.]

Thank you for resubmitting your work entitled "Population coupling predicts the plasticity of stimulus responses in cortical circuits" for further consideration by *eLife*. Your revised article has been evaluated by Ronald Calabrese (Senior Editor) and a Reviewing Editor.

The manuscript has been improved but there are some remaining issues that need to be addressed before acceptance, as outlined below:

Please make the following very minor but important edits directed by the reviewer comments provided. These should be accomplished rapidly so as not to hold-up publication and will require only the Reviewing Editor to review.

Reviewer #1:

Figure 1—figure supplement 1 discussion: Why change the parameter for the synaptic scaling rate from zeta to eta? Does zeta refer to something else?

Figure 3E: missing x axis ticks for static gratings.

Throughout, the figures have a variety of font sizes for the tick labels, axis labels, and legends. Making these uniform, within and across figures, would make them easier to read.

Reviewer #2:

In their revised paper, Sweeney and Clopath have addressed most of the concerns raised by myself and the other reviewers quite well. In my opinion, the paper is very close to being ready for publication. I have one final quibble related to the first major concern I raised in my initial review.

In my initial review, I noted that there is an alternative hypothesis based on normative models which posits that neurons with strong population coupling should be more plastic because they have a greater impact on the network, and thus are more important for any performance gradients. But. this hypothesis also posits that neurons that inhibit other neurons a lot, thus which may have extremely low population coupling, would also be more plastic.

The authors have almost satisfied my concerns with their response. I appreciate their attempt at testing whether this prediction holds in the data. However, their test is ill-formed. They test this hypothesis by assigning a semi-arbitrary threshold for low (< 0.2) vs. medium (>=0.2, <=0.4) vs. high (>0.4) population coupling, and then testing whether neurons in the Allen data with low population coupling show bigger orientation changes than neurons with medium orientation changes. They report that the tests were non-significant, and conclude that the data doesn't support this hypothesis.

This test rests on the idea that we can group the neurons into these three bins and hope to test this alternative hypothesis. Yet, we do not really know how the scale of the population coupling metric used here relates to real impact on network activity in the brain. For example, it could be that neurons with a population coupling metric of 0.3 actually have a fairly big impact on the network, so they should be in the high group. Essentially, we have no principled way to judge what the threshold for these groups should be. Indeed, I am not surprised that the tests came out as non-significant, because it is clear that the Δ ORI pref begins to increase after a level of around 0.25 in population coupling.

As it stands, by my eye, the data in Figure 3E still suggests that there is an initial decrease in Δ ORI pref as we move from really low population coupling, then an increase again as we move further out. The test given does not convince me otherwise. Really, if the goal were to convince me that this alternative hypothesis cannot account for the data as well as their model, the authors would need to determine whether a linear model with nothing but a monotonic increase in Δ ORI pref accounts for the data better than a non-linear or bi-linear model that include an initial decrease. My bet is that the latter model would be better at fitting the data, even after consideration of increased parameters (e.g. via the AIC or something).

However, I don't need to see this analysis per se, and I don't intend to hold up publication of this paper based on my personal interpretation of the data. What I would like to see is simply a change in the text that is now given in the Discussion. The authors write:

"Thus, a gradient-based account would predict that there should be a U-shaped curve relating population coupling to plasticity. Although we did not observe such a relationship, this may be a fruitful avenue of enquiry for future experiments."

I think that, to be more honest, this should read something more like:

"Thus, a gradient-based account would predict that there should be a U-shaped curve relating population coupling to plasticity. Our data showed may show some initial decrease in plasticity as population coupling increases (Figure 3F), but these questions outstrip the focus of this work. Consideration of whether alternative models based on gradients could explain this data may nonetheless be a fruitful avenue of enquiry for future experiments."

---

## [Author Response]

[Editors’ note: the authors resubmitted a revised version of the paper for consideration. What follows is the authors’ response to the first round of review.]

A new submission of a thoroughly reworked paper that addresses these concerns and additional concerns in original reviews could be viewed favorably at eLife. The Reviewing Editor would particularly refer the authors to concerns by reviewer #1 "I would have expected an associated Figure 3E and F from the model so that a better comparison could be made and we could understanding how much the α's need to vary."

We have now included a more direct comparison between the data and our model, by including data from the network simulations within Figure 3F. Both the network simulation and experimental data are consistent with a link between population coupling and fluctuations in selectivity.

and "The relation between learning diversity and population coupling is interesting – but outside of the initial observation in the model there is no real theory/understanding for this relation being offered",

We have also included an exploration of how much the α’s (learning rate) need to vary in order to match the distribution of population coupling observed experimentally (Figure 3—figure supplement 2, distribution shown in Figure 3C).

and from reviewer #3 "They begin with diverse learning rates and then examine the effect on population coupling. But, what if, functionally, population coupling should determine learning rate, as stipulated by gradient descent?"

We thought that this was a very interesting hypothesis, and tested it further in the data (described below). We found no strong statistical evidence for this in the current data, but now discuss it in the manuscript.

Reviewer #1:The paper by Sweeny and Clopath investigates how a diversity of plasticity timescales allows for a diversity of neuronal coupling to population activity. Overall, the paper investigates an interesting effect. However, I feel that the paper falls short in several areas. I explain these below.1) While the comparison to the Allen dataset is inspired, the end result is somewhat disappointing. The relation between change in preferred orientation and population coupling shown in Figure 3E and F are quite weak (I realize that it is statistically significant). Further, there is no attempt to relate the weak trend in the data to the distribution of learning timescales that would be required to produce it. This is likely because the model is simplified and there is not a real sense of continuous tuning (only four orientations were shown). In this way the model prediction is not really tight. I would have expected an associated Figure 3E and F from the model so that a better comparison could be made and we could understanding how much the α's need to vary.

We also now calculate the population coupling ratio for neurons with low/high fluctuations in the model, and show them alongside the data in Figure 3F. The trend is similar to that seen in the data, i.e. the ratio > 1 across 15 network simulations. Neurons with high fluctuations in their selectivity have a higher population coupling than neurons with low fluctuations. We define low/high fluctuations by whether there are < or > the median fluctuation.

Although there are indeed 4 orientations in the Figure 1 model, there are 8 orientations in the Figure 2 models, which is a more detailed model. While conducting our initial network simulations we also explored a wider range of orientations (up to 32), and simulations in which orientation tuning is randomly generated from a continuous distribution. We found that the trend between population coupling and change in selectivity was conserved across these simulations, but chose to use simulations with 8 orientations in our extended analysis, as this number of orientations are more consistent with the Brain Observatory data. In addition, we found that we need a large number of neurons per orientation to gather enough observations about changes in selectivity to use in our analyses (i.e. if we used 32 orientations we would need to simulate with 4x as many neurons, which was infeasible given our computational resources).

2) The relation between learning diversity and population coupling is interesting – but outside of the initial observation in the model there is no real theory/understanding for this relation being offered. Is there a simple mean field style theory that can be put forth? Can other sources of heterogeneity (say single neuron baseline firing rate r_0_) create a similar relation? Why is there a spread of population coupling in the uniform α case? Answers to these questions could help build a deeper understanding of the key result. As it stands the paper simply shows simulation results and gives a loose connection between synaptic variability, population coupling, and response tuning. I expect more in a high profile publication.

The theory for understanding the link between population coupling and synaptic input diversity has already been proposed by Okun et al., which theoretically and empirically link population coupling to specific versus non-specific synaptic connections (and discuss how factors such as baseline firing rate doesn’t contribute to this effect). We build upon this, proposing a mechanism that gives rise to the diverse ratios of specific/non-specific connections. Of course, there are many different plasticity rules which could achieve this, but a key factor is that – assuming some form of Hebbian plasticity – some neurons are more sensitive to transiently correlated ‘population’ activity than others, and develop stronger coupling to it. For us, this is expressed as the learning rate (see Figure 1—figure supplement 1).

This can be expressed by the terms in equation 1, below. For specific weights (W_spec_), corr(stim) is high, meaning that the W_spec_ tends to be large. For non-specific weights, corr(stim) is ~ 0. From the terms in equation 1, we can see that a large α, large corr(noise), and small eta, can all contribute to a large W_nonspec_

dW = α*(corr(stim)+corr(noise)) – eta*(W_total_-W_target_) Equation 1

This is now discussed alongside Figure 1—figure supplement 1.

We also investigated how other forms of heterogeneity affects the distribution of population coupling, e.g. the single neuron baseline firing rate r_0_, as suggested by the reviewer. We found that introducing a uniform distribution of r_0_, along a range which we also varied, did not introduce substantial changes in the distribution of population coupling (shown in Author response image 1).

**Author response image 1. respfig1:** Distribution of population coupling in network simulations as we vary the range of the uniform baseline firing rate distribution, from 1 to 15.

Reviewer #2:Sweeney and Clopath report simulations and analysis of in vivo calcium imaging data suggesting that neurons with faster learning rates will exhibit higher population coupling, consistent with the idea of functional network architecture composed of slowly-changing connectivity between stably tuned neurons and quickly changing connectivity between unstably tuned neurons.The authors begin by studying networks where they train a low-dimensional structure into the excitatory weights (four-dimensional in Figure 1, and I would guess 8-dimensional in Figure 2 due to the 8 orientations of receptive field), with one global inhibitory neuron. Population coupling describes a one-dimensional correlation structure – is this an appropriate description for these network, or are the activity four or eight dimensional? How do the correlations within- and between the trained structures relate to the population coupling, and how does the timescale of a neuron's plasticity relate to its coupling to co-stimulated neurons vs non-co-stimulated neurons?

Due to the relatively high amount of injected noise and recurrent connections within the network, the network activity does not tend to be low-dimensional. This is demonstrated in the plots in Author response image 2 showing the pairwise correlation matrix (left) of the activity of neurons in a network with 8 orientations, and where there is no discernible correlation structure. The distribution of pairwise correlation coefficients is also a broad and unimodal (right). The relation between the timescale of a neurons’ plasticity and its coupling to co-stimulated vs non-co-stimulated neurons is shown in Figure 1E (the co-stimulated neurons are ‘specific connections’ and the non co-stimulated neurons are ‘non-specific).

**Author response image 2. respfig2:** Pairwise correlations.

What is the relationship between the timescale of the associations trained in Figure 4, and do the results depend on the relationship between that timescale of stimulation and the fast/slow timescales of plasticity?

The timescale of the change in stimulation is 25 seconds, and is slower than the timescale of both slow and fast plasticity. The change in stimulation needs to be slow enough so that learning the associated stimulus occurs, so that we can demonstrate that neurons with fast plasticity exhibit earlier and stronger associative learning. Although the associated stimulus switches before the neurons with slow plasticity complete learning, we can still see the cumulative effect of the associative learning paradigm in Figure 4A.

I applaud the authors for seeking a direct experimental test of their model's results. The results of that test (Figure 3E) seem, however, fairly weak. More importantly, it seems that the basic characterization of tuning stability in this data has been skipped over, which makes it difficult to interpret these results.The trial-by-trial responses to gratings (at least in the Allen Brain Observatory dataset) exhibit significant variability, and the tuning curve is often a poor predictor of the responses. This requires, I think, a stronger analysis to test the authors' prediction.a) First off, how stable is the tuning? Is orientation selectivity stable? If there are neurons that are tuned only in the early or the late epoch, how does that match with the model?

We quantify the stability of tuning by looking at the reliability of orientation responses (Figure 3, Figure 3—figure supplement 1). The reliability is a measure provided in the Allen Brain Observatory API, and is the mean trial-to-trial correlation of the dF/F calcium fluorescence traces during the neurons preferred stimulus presentation. This seemed to us a good measure for assessing the stability of tuning. There is a relatively broad distribution of reliabilities across neurons, and the distribution differs somewhat for the drifting grating and static grating experiments, with neurons having more reliable responses in the drifting grating experiments. This is encouraging, as it is also the stimulus set in which we see the strongest link between population coupling and plasticity of stimulus selectivity. So, we don’t think we are seeing these effects because the neurons are simply very unreliable. To test this further, we reanalyse the effects using only neurons with high reliability across the experiment, and find that the correlation between population coupling and plasticity is in fact stronger for these neurons, across both static and drifting grating stimulus sets.

b)In Figure 3B, it seems that the neurons with the largest responses are the most population coupled, which makes sense given the definition of population coupling. How does this contribute to the neurons' orientation selectivity and relate to tuning stability?

We explore this relationship – and other relevant measures – in Figure 3—figure supplement 4. We found that there is no statistically significant relationship between mean response and tuning stability (Figure 3—figure supplement 1B), but that peak response and tuning stability are correlated (Figure 3—figure supplement 4C). We found that the relationship between the plasticity of tuning and population coupling remained when controlling for peak responses as a potential confounding factor (Figure 3—figure supplement 4F).

c) In neurons that are significantly tuned at the early and late epochs, is their tuning different? This is close to what the authors asked, but I think it could be made more rigorous – one idea is by a statistical test asking whether the distribution of responses to the initially preferred stimulus is distinguishable between the early and late grating epochs. (And correspondingly for the preferred stimulus of the late epoch.)

While the Brain Observatory API does not provide a way to easily extract the full distribution of responses divided between early and late epochs, we can measure the distributions of the change in mean and variance of the responses to the initially preferred stimulus across the early and late grating epochs. We plot the distribution of signal to noise ratio (SNR), define as |mean(early)-mean(late)|/mean(std(early)+std(late)). In the plot in Author response image 3, the changes in early and late responses are either for responses to the initially preferred stimulus (early), the preferred stimulus of the late epoch (late) if there is a change in preferred stimulus, or to a random non-preferred stimulus (random). As may be expected from previous observations about the distribution of fluctuations in stimulus selectivity (Figure 3D), there is a relatively broad range of changes in the response distribution. The SRN distribution of response changes does not seem dependent on whether they are responses to a preferred stimulus or not, although the SNR distribution is more similar for responses to either of the preferred stimuli (KS statistic =.035), compared with responses to a non-preferred stimulus (KS statistic =0.11 (late) and 0.14 (early)).

**Author response image 3. respfig3:** Distribution of SNR.

d)In Figure 3B, it seems that the neurons with the largest responses are the most population coupled, which makes sense given the definition of population coupling. Does this hold up with extracted events? How does this contribute to the neurons' orientation selectivity and relate to tuning stability?

The extracted events detected by the brain observatory API doesn’t give us much information, as there aren’t enough events per recording to get good estimates of the population coupling. Recall that it is difficult even to get a decent measure of population coupling even with the df/F fluorescence traces, and we had to discard a substantial number of experiments that didn’t meet our criteria. The number of events is anti-correlated with population coupling, and with orientation preference stability, and uncorrelated with OSI (see Author response image 4).

**Author response image 4. respfig4:** Population Coupling.

e) How do these results relate to the degree of neurons' tuning and the reliability of their responses? Do strongly tuned neurons have more stable stimulus preferences? Do neurons with lower trial-to-trial variability have more stable stimulus preferences?

See above – We measure the reliability of responses across neurons, and find that the results hold, and are stronger, in neurons with high reliability. This is shown in now in Figure 3 and in Figure 3—figure supplement 1.

f) The authors refer only to orientation tuning. For the drifting gratings, what about direction? (And what about spatial or temporal frequencies?)

The direction tuning of drifting grating is also included in our analysis (black versus gray stacks in the histograms). The original Figure 3 considered both orientation and direction tuning together, we now show the analysis and results separately.

We have now investigated spatial and temporal frequencies. The same effect is present in both these case (Figure 3—figure supplement 3). Note also that orientation tuning exhibits a weaker effect than direction tuning, consistent with a static/drifting grating difference. It’s possible that this is just due to them being different experiments, or due to there being more plasticity evoked by drifting gratings versus static gratings.

g) The description of which data from the Allen Brain Observatory were used seems incomplete. How were experiments chosen to include; which Cre lines, layers, and areas were the data used from? Do the results differ across those dimensions? That dataset currently includes a lot more than the 112 experiments with gratings included.

We now include a list of the ids of the experiments we used to perform the analysis, as a table available at 10.6084/m9.figshare.11837406. Note also that the analysis was conducted with the first round of datasets released, there have been further releases of datasets since then. We selected datasets from primary visual area (visp) and with line Ai93. The details and list of experiments is now included in our Materials and methods section. Conducting an analysis of how the results differ across layer/areas/Cre lines is beyond the scope of our study, but can be performed using the analysis scripts that we will provide.

h) The observatory API also included extracted events, to account for the autocorrelation time of the fluorescence. Do the results depend on the use of df/f compared to events?

As above, extracted events aren’t useful for us in this analysis – there are not enough samples per recording session.

Reviewer #3:[…]I have two related concerns about the findings in the paper, though, which I would encourage the authors to consider. The root of my concerns is the lack of a normative framework for understanding learning in both the models and the data. The authors begin with diverse learning rates as a potential phenomenon, then attempt to make predictions and explore functional implications. I wonder whether flipping this might not have been a more fruitful approach for explaining the experimental data. That is, the authors could have started with a question of what functional problem is to be solved, then showed that diverse learning rates could solve it, and then confirmed the phenomenon in the data. There are two reasons why this more normative approach may be better at explaining the data:1) An alternative account for the phenomenon the authors observe in the Allen data can be found in normative models. Consider a recurrent neural network trained with gradient descent to perform some function. Regardless of the function being learned, on average, neurons with greater impact on the rest of the network will have larger synaptic updates than other neurons. This is because the gradient of any function with respect to a neuron will depend on its ability to influence future activity, and neurons that contribute more to the gradient will be updated more. Now, neurons with a strong tendency to activate other neurons will have high levels of population coupling, while neurons with a tendency to inhibit other neurons (disynaptically, since these are excitatory neurons) will have low levels of population coding. Neurons with limited impact will have middle levels of population coupling. Thus, a gradient based account would predict that there should be a U-shaped curve relating population coupling to plasticity. To my eye, that is precisely what the Allen data actually shows (Figure 3E). Thus, I worry that the authors have it backwards. They begin with diverse learning rates and then examine the effect on population coupling. But, what if, functionally, population coupling should determine learning rate, as stipulated by gradient descent? Now, to be clear, I do not think the authors have to actually address this hypothesis I put forward with new simulations. Per the eLife philosophy, I would like to improve this paper, not demand the paper I may have been inclined to write myself. So, I don't think new simulations are needed. However, given the slightly questionable fit of their model hypothesis to the data shown in Figure 3E, I think that this alternative hypothesis should at least be mentioned in the Discussion.

This is certainly an interesting hypothesis, and we tested this out on the Allen Brain Institute data. Specifically, we tested whether neurons with low population coupling were also more plastic than neurons with average population coupling (i.e. similarly to neurons with high population coupling). To do this, we split the neurons in three groups (low, moderate, high) according to their population coupling. We then tested whether the change in stimulus selectivity was significantly different across neurons in these low/moderate/high groups. The hypothesis – that neurons with low population coupling have higher plasticity of stimulus selectivity than neurons with moderate population coupling, was rejected in all four stimulus types (orientation, direction, spatial frequency, and temporal frequency). We discuss this in the text.

We did not find such a relationship, finding that (for orientation, direction, and spatial frequency) the plasticity of stimulus selectivity was higher amongst neurons with high population coupling versus low population coupling, and amongst neurons with high population coupling versus moderate population coupling, but not in any other pairwise comparison (Tukey’s multiple comparison test, results below). There were no statistically significant differences for temporal frequency.

Population coupling is defined as low if < 0.2 (n=230[DG], 602 [SG]), high if > 0.4 (n=193 [DG], 2643 [SG]), and moderate otherwise (n = 717 [DG], 514 [SG]). DG=drifting gratings, SG=static gratings.

Spatial frequency tuning:

Multiple Comparison of Means – Tukey HSD,FWER=0.05

group1group2meandifflowerupperrejectLowMedium-0.0305-0.13910.078FalseLowHigh0.2150.07170.3584TrueMediumHigh0.24560.12980.3613True

Orientation tuning:

Multiple Comparison of Means – Tukey HSD,FWER=0.05

group1group2meandifflowerupperrejectLowMedium1.5271-4.55667.6108FalseLowHigh11.21592.531719.9TrueMediumHigh9.68882.327817.0499True

Temporal frequency tuning:

Multiple Comparison of Means – Tukey HSD,FWER=0.05

group1group2meandifflowerupperrejectLowMedium-0.0335-0.16880.1019FalseLowHigh0.0578-0.12680.2425FalseMediumHigh0.0913-0.06820.2508False

Direction tuning:

Multiple Comparison of Means – Tukey HSD,FWER=0.05

group1group2meandifflowerupperrejectLowMedium8.9885-4.95622.9331FalseLowHigh35.612817.649753.5758TrueMediumHigh26.624211.70241.5465True

It is possible that further experiments could uncover such a relationship however, so we have included a discussion of this hypothesis in the manuscript.

2) The purpose of the simulations in Figure 4 were a bit mysterious to me. The authors appeared to make fairly strong assumptions about what would be useful to a network, e.g. that it would be good to show increased selectivity to some external stimulus, or that information should be as preserved as possible. But, why not start with the more obvious normative framework for multiple learning rates, and one which the authors identify in the Discussion, i.e. the need to avoid catastrophic forgetting? This is sort of demonstrated in Figure 4, but in an indirect fashion that relies on the assumption that selectivity is a good performance metric. Why not actually use the networks to perform various tasks, and show that the networks are better at avoiding catastrophic forgetting of the tasks when diverse learning rates are present? Even a simple task would be easier to understand the implications, than the selectivity measurements provided by the authors, in my opinion. But, again, I don't actually think new simulations would be required, per se (though they could help). Nonetheless, I think a more intuitive account for why selectivity is the right metric to use would be helpful.

The choice of selectivity as a metric was primarily so that we could perform a like-for-like comparison with what could be measured from the empirical data, i.e. stimulus selectivity. We fully agree with the reviewer that selectivity as a metric doesn’t capture everything, and that there are alternative approaches which may capture more complex stimulus representations, and the plasticity of these representations. Indeed, it would be fascinating to conduct an analysis of how plasticity rate relates to neurons involvement in low-dimensional latent dynamics, similar to that found by Gallego et al. recently, but this is beyond the scope of our current study. We now discuss this in the manuscript in subsection “A plasticity-coupling link in vivo”. Note also that we do measure task performance in Figure 4E, where the task is to decode the identity of pairs of stimuli presented simultaneously to the network. As the reviewer hints at, this task can already be thought of as an indirect test of overcoming catastrophic forgetting, since the stimulus protocol throughout development involves the continual presentation of stimuli throughout synaptic plasticity. Hence the network composed entirely of neurons with fast learning rates performs poorly in this task.

[Editors’ note: what follows is the authors’ response to the second round of review.]

Please make the following very minor but important edits directed by the reviewer comments provided. These should be accomplished rapidly so as not to hold-up publication and will require only the Reviewing Editor to review.Reviewer #1:Figure 1—figure supplement 1 discussion: Why change the parameter for the synaptic scaling rate from zeta to eta? Does zeta refer to something else?

We corrected the typo.

Figure 3E: missing x axis ticks for static gratings.

We added that.

Throughout, the figures have a variety of font sizes for the tick labels, axis labels, and legends. Making these uniform, within and across figures, would make them easier to read.

We tried our best to uniform the figures.

Reviewer #2:[…]However, I don't need to see this analysis per se, and I don't intend to hold up publication of this paper based on my personal interpretation of the data. What I would like to see is simply a change in the text that is now given in the Discussion. The authors write:"Thus, a gradient-based account would predict that there should be a U-shaped curve relating population coupling to plasticity. Although we did not observe such a relationship, this may be a fruitful avenue of enquiry for future experiments."I think that, to be more honest, this should read something more like:"Thus, a gradient-based account would predict that there should be a U-shaped curve relating population coupling to plasticity. Our data showed may show some initial decrease in plasticity as population coupling increases (Figure 3F), but these questions outstrip the focus of this work. Consideration of whether alternative models based on gradients could explain this data may nonetheless be a fruitful avenue of enquiry for future experiments."

We did change the paragraph as suggested.